# QERA: AN ANALYTICAL FRAMEWORK FOR QUANTIZATION ERROR RECONSTRUCTION

**Cheng Zhang, Jeffrey T. H. Wong, Can Xiao, George A. Constantinides & Yiren Zhao**
Department of Electrical and Electronic Engineering
Imperial College London
London, UK
{cheng.zhang122,tsz.wong20,can.xiao22,g.constantinides,a.zhao}@imperial.ac.uk

## ABSTRACT

The growing number of parameters and computational demands of large language models (LLMs) present significant challenges for their efficient deployment. Recently, there is an increasing interest in quantizing weights to extremely low precision while offsetting the resulting error with low-rank, high-precision error reconstruction terms. The combination of quantization and low-rank approximation is now popular in both adapter-based, parameter-efficient fine-tuning methods such as LoftQ (Li et al., 2023) and low-precision inference techniques including ZeroQuant-V2 (Yao et al., 2023). Usually, the low-rank terms are calculated via the singular value decomposition (SVD) of the weight quantization error, minimizing the Frobenius and spectral norms of the weight approximation error. Recent methods like LQ-LoRA (Guo et al., 2023) and LQER (Zhang et al., 2024a) introduced hand-crafted heuristics to minimize errors in layer outputs (activations) rather than weights, resulting improved quantization results. However, these heuristic-based methods lack an analytical solution to guide the design of quantization error reconstruction terms. In this paper, we revisit this problem and formulate an analytical framework, named Quantization Error Reconstruction Analysis (QERA), and offer a closed-form solution to the problem. We show QERA benefits both existing low-precision fine-tuning and inference methods – QERA achieves a fine-tuned accuracy gain for $\Delta_{\text{acc}} = 6.05\%$ of 2-bit RoBERTa-base on GLUE compared to LoftQ; and obtains $\Delta_{\text{acc}} = 2.97\%$ higher post-training quantization accuracy of 4-bit Llama-3.1-70B compared to ZeroQuant-V2 and $\Delta_{\text{ppl}} = -0.28$ lower perplexity on WikiText2 compared to LQER. We open-source our code and models at github.com/ChengZhang-98/QERA.

## 1 INTRODUCTION

The demand for efficient deployment of large language models (LLMs) has been increasing (Faiz et al., 2023). LLMs now typically contain billions of parameters (Kaplan et al., 2020; Dubey et al., 2024), making their fine-tuning and inference computationally expensive and resource-intensive (Ding et al., 2023). To address these challenges, there has been a surge of interest in building efficient fine-tuning and inference methods. One popular formulation is to apply a low-rank term to reconstruct the error after quantization. Given a linear layer $\boldsymbol{y} = \boldsymbol{xW}$, the weight matrix $\boldsymbol{W} \in \mathbb{R}^{m \times n}$ is quantized to $\widetilde{\boldsymbol{W}}$, and we rewrite $\boldsymbol{W} \approx \widetilde{\boldsymbol{W}} + \boldsymbol{A}_k \boldsymbol{B}_k$ such that both $\boldsymbol{A}_k$ and $\boldsymbol{B}_k$ are low-rank yet high-precision terms with rank $k \ll \min(m, n)$.

We call the problem of finding the optimal $\boldsymbol{A}_k$ and $\boldsymbol{B}_k$ *quantization error reconstruction*. Interestingly, this problem has, coincidentally, seen widespread application in two actively researched areas: quantized parameter-efficient fine-tuning (QPEFT) and post-training quantization (PTQ) for model inference. QPEFT refers to fine-tuning techniques that adapt LLMs to specific tasks by quantizing pretrained weights and updating only a small number of extra parameters, hence significantly reducing memory requirements and training time, such as QLoRA (Guo et al., 2023). On the other side, PTQ is a training-free method that reduces the model size and may accelerate the forward pass if the underlying hardware supports it. Recently, researchers combined PTQ with quantization error reconstruction (Yao et al., 2023; Liu et al., 2023a; Zhang et al., 2024a) to further reduce weight

precision. Works such as ZeroQuant-V2 (Yao et al., 2023) and LQER (Zhang et al., 2024a) have shown that adding a high-precision low-rank component, as low as 8 or 32, can recover considerable model performance for 3- or 4-bit weight quantization.

Although both the QPEFT and PTQ methods have demonstrated substantial performance improvements in lowering the computational overhead of LLMs, a theoretical analysis of quantization error reconstruction is lacking. Usually, $\boldsymbol{A}_k$ and $\boldsymbol{B}_k$ are calculated by applying truncated singular value decomposition (SVD) to the weight quantization error $(\boldsymbol{W} - \widetilde{\boldsymbol{W}})$, minimizing the Frobenius and spectral norms of the *weight approximation error*. However, recent work on activation-aware quantization and knowledge distillation implies that minimizing *layer output error* may lead to a greater performance gain than minimizing weight approximation error (Lin et al., 2024; Liu et al., 2023a; Shao et al., 2023).

Besides the unsettled minimization objective, it has remained unclear whether there exists a theoretically optimal solution for the values of $\boldsymbol{A}_k$ and $\boldsymbol{B}_k$, and if so, how one can solve for it. A better initialization or theoretically grounded initialization of $\boldsymbol{A}_k$ and $\boldsymbol{B}_k$ brings direct benefits for both QPEFT and PTQ. In QPEFT, the initialization of LoRA (Hu et al., 2021), which uses element-wise Gaussian random values for $\boldsymbol{A}_k$ and zeros for $\boldsymbol{B}_k$, struggles under aggressive quantization since the quantization error can derail fine-tuning. In PTQ, the quantized model performance is based on the computation of the low-rank terms, given a specific quantization function $\mathrm{q}(\cdot)$ and rank $k$.

In this paper, we aim to provide an analytical framework for the *quantization error reconstruction* problem. To demonstrate the effectiveness of our theoretical framework, we further apply our analytical solutions to state-of-the-art QPEFT and PTQ methods and show the significant performance improvements under the same computational budget. Specifically, our contributions are as follows:

- We show that the commonly used objective for solving the *quantization error reconstruction* problem in prior work , *i.e.*, minimizing the weight approximation error (*e.g.*, $||\boldsymbol{W} - \widetilde{\boldsymbol{W}}||_p$), does not guarantee a reduced *model output error*. Instead, we show that minimizing the layer output error (*e.g.*, $||\boldsymbol{y} - \widetilde{\boldsymbol{y}}||_p$) is closely related to minimizing the model output error.

- We derive the analytical solution to the low-rank terms $\boldsymbol{A}_k$ and $\boldsymbol{B}_k$ by minimizing the layer output error. We demonstrate that under a statistical assumption, this solution can be found in a particularly computationally efficient manner, also explaining the success of LQER.

- We empirically demonstrate the effectiveness of our solutions by applying them to state-of-the-art QPEFT and PTQ methods. Our analytical framework, QERA, significantly improves the performance of these methods. For example, QERA achieves $\Delta_{\mathrm{acc}} = 6.05\%$ higher accuracy of 2-bit RoBERTa-base on GLUE compared to LoftQ, improving the fine-tuning accuracy and efficiency. Moreover, QERA obtains $\Delta_{\mathrm{acc}} = 2.97\%$ higher accuracy than ZeroQuant-V2, when quantizing LLaMA-3-70B to 4 bits, averaged across six tasks. This narrows the model performance gap between error-reconstruction-based post-training quantization and full-precision models.

## 2 RELATED WORK

In this section, we review the existing methods that combine weight quantization and low-rank error reconstruction. These methods can be roughly categorized into two groups based on their applications: QPEFT for training and PTQ for inference.

**LoRA and QPEFT**   LoRA Hu et al. (2021) is a representative PEFT method that introduces trainable low-rank terms to adapt the model to a specific task. Take a linear layer as an example,

$$\boldsymbol{y} = \boldsymbol{x}(\boldsymbol{W} + \boldsymbol{A}_k \boldsymbol{B}_k) \tag{1}$$

where $\boldsymbol{W} \in \mathbb{R}^{m \times n}$ is the pretrained weight matrix, row vector $\boldsymbol{x} \in \mathbb{R}^m$ and $\boldsymbol{y} \in \mathbb{R}^n$ are the input and output, and $\boldsymbol{A}_k \in \mathbb{R}^{m \times k}$ and $\boldsymbol{B}_k \in \mathbb{R}^{k \times n}$ are trainable low-rank matrices ("adapter") with rank $k \ll \min(m, n)$. During fine-tuning, the pretrained $\boldsymbol{W}$ is frozen and only the adapter $\boldsymbol{A}_k$ and $\boldsymbol{B}_k$ are updated. To make the adapted layer's output match the original one at the start of fine-tuning, LoRA initializes $\boldsymbol{A}_k$ with Gaussian random values and $\boldsymbol{B}_k$ with zeros. Once the fine-tuning is completed, the adapter is merged into the pre-trained weights.

QLoRA (Guo et al., 2023) extends LoRA by quantizing the pretrained weights stored in GPU memory to reduce memory footprint.

$$\boldsymbol{W}_q = \mathrm{q}(\boldsymbol{W}) \tag{2}$$

One difference between QLoRA and LoRA is that during fine-tuning, $\boldsymbol{W}_q$ needs to be dequantized before involved into matrix multiplications:

$$\widetilde{\boldsymbol{W}} = \mathrm{dq}(\boldsymbol{W}_q), \ \boldsymbol{y} = \boldsymbol{x}(\widetilde{\boldsymbol{W}} + \boldsymbol{A}_k\boldsymbol{B}_k) \tag{3}$$

where $\mathrm{dq}(\cdot)$ is the dequantization function. QLoRA introduces weight quantization error ($\boldsymbol{W} - \widetilde{\boldsymbol{W}}$), shifting the starting point of fine-tuning. To address this problem, LoftQ (Li et al., 2023) initializes the adapter using the SVD-based low-rank approximation of ($\boldsymbol{W} - \widetilde{\boldsymbol{W}}$) to reduce the weight approximation error:

$$\underset{\boldsymbol{A}_k, \boldsymbol{B}_k}{\arg\min} ||\boldsymbol{W} - \widetilde{\boldsymbol{W}} - \boldsymbol{A}_k\boldsymbol{B}_k||_F \tag{4}$$

Specifically, LoftQ uses a heuristic-based algorithm to iteratively update the quantized weights and the adapter (Algorithm 1 in the Appendix). Their experiments show that a larger number of iterations leads to a smaller weight error.

LQ-LoRA (Guo et al., 2023) also adopts LoftQ's iterative method but keeps track of a scaled variant of the objective, $\arg\min_{\boldsymbol{A}_k, \boldsymbol{B}_k} ||\boldsymbol{D}_{\mathrm{row}}(\boldsymbol{W} - \widetilde{\boldsymbol{W}} - \boldsymbol{A}_k\boldsymbol{B}_k)\boldsymbol{D}_{\mathrm{col}}||_F$, where $\boldsymbol{D}_{\mathrm{row}}$ and $\boldsymbol{D}_{\mathrm{col}}$ are heuristic homogenous row/column matrices from activation statistics. LQ-LoRA exits the iteration when the scaled objective function stops decreasing due to the lack of a theoretical justification for LoftQ.

**Quantization Error Reconstruction for PTQ**  Similar to the forward pass of fine-tuning in QLoRA, there are also PTQ methods that quantize the pretrained weights to low-precision formats and recover the model performance with additional low-rank terms. With a small enough rank $k$, the additional computation introduced is negligible. Note that unlike QPEFT which can utilize fine-tuning to correct the quantization error, PTQ methods aim to recover the model performance as much as possible without any training.

ZeroQuant-V2 (Yao et al., 2023) is the earliest weight-only quantization method introducing low-rank quantization error reconstruction to the PTQ problem. They apply SVD to the weight quantization error ($\boldsymbol{W} - \widetilde{\boldsymbol{W}}$) to calculate $\boldsymbol{A}_k$ and $\boldsymbol{B}_k$ (equivalent to LoftQ with one iteration). Combining low-rank terms and fine-grained quantization, ZeroQuant-V2 recovers the performance of 4-bit LLMs to a level comparable to 8-bit.

Recent quantization works have shown that activation statistics play a crucial role in weight-only LLM quantization (Liu et al., 2023b; Lin et al., 2024). QLLM (Liu et al., 2023a) trains the low-rank terms using gradient descent with a loss function that minimizes the output error of the attention layer. LQER (Zhang et al., 2024a) applies an activation-induced heuristic scale matrix $\boldsymbol{S}$ to the quantization error before calculating SVD, $\boldsymbol{U}\boldsymbol{\Sigma}\boldsymbol{V}^T = \mathrm{SVD}(\boldsymbol{S}(\boldsymbol{W} - \widetilde{\boldsymbol{W}}))$, and assigns $\boldsymbol{A}_k := \boldsymbol{S}^{-1}\boldsymbol{U}_{:,:k}$, $\boldsymbol{B}_k := \boldsymbol{\Sigma}_{:k,:k}\boldsymbol{V}_{:k,:}^T$ (Refer to Algorithm 2 in the Appendix). LQER achieves significant improvement over ZeroQuant-V2 and observes that in some layers singular values are shaped toward a more desirable distribution where singular values decay faster. Note that ZeroQuant-V2 can also be considered as a special case where $\boldsymbol{S}$ in LQER is an identity matrix. To our knowledge, CALDERA (Saha et al., 2024) is the concurrent work close to ours. CALDERA focuses on a different problem setup to find optimal $\widetilde{W}, A_k, B_k$ all in low-precision formats that minimizes output error, with a lemma agreeing with our exact solution. We elaborate the connection and difference between CALDERA and QERA in Appendix A.3.

In summary, to solve the quantization error reconstruction (QER) problem, most of existing methods target the minimization of the weight approximation error. Several recent works such as LQ-LoRA, QLLM, and LQER introduce activation-induced heuristics to the calculation of adapters/low-rank terms, but **a justification for the optimization objective and the corresponding analytical framework are still missing**.

## 3  OUR ANALYTICAL FRAMEWORK

In this section, we formulate the optimization objective of quantization error reconstruction and derive the analytical solution to the low-rank term $\boldsymbol{C}_k := \boldsymbol{A}_k\boldsymbol{B}_k$.

## 3.1 PROBLEM STATEMENT

Given a pretrained linear layer $\boldsymbol{y} = \boldsymbol{xW}$ with input vector $\boldsymbol{x} \in \mathbb{R}^m$, output vector $\boldsymbol{y} \in \mathbb{R}^n$, and weight matrix $\boldsymbol{W} \in \mathbb{R}^{m \times n}$, our aim is to approximate it with a high-rank low-precision $\widetilde{\boldsymbol{W}}$ and a low-rank high-precision term $\boldsymbol{C}_k \in \mathbb{R}^{m \times n}$ with rank $k \ll \min(m, n)$.

$$\widetilde{\boldsymbol{y}} = \boldsymbol{x}(\widetilde{\boldsymbol{W}} + \boldsymbol{C}_k) \tag{5}$$

This raises the question of the actual optimization target: Should we minimize the weight reconstruction error $||\boldsymbol{W} - \widetilde{\boldsymbol{W}}||_F$ or the output reconstruction error $||\boldsymbol{y} - \widetilde{\boldsymbol{y}}||_2$? We separate these two problems and introduce them formally below.

**Problem 1** (Minimization of weight error). For a pretrained linear layer $\boldsymbol{y} = \boldsymbol{xW}$ and its approximated form $\widetilde{\boldsymbol{y}} = \boldsymbol{x}(\widetilde{\boldsymbol{W}} + \boldsymbol{C}_k)$, reconstructing the quantization error by minimizing weight approximation error has the following objective:

$$\arg\min_{\boldsymbol{C}_k} ||\boldsymbol{W} - \widetilde{\boldsymbol{W}} - \boldsymbol{C}_k||_F \tag{6}$$

where $|| \cdot ||_F$ denotes the Frobenius norm.

**Solution to Problem 1**. From the Eckart-Young-Mirsky theorem (Eckart & Young, 1936), the optimal solution to Problem 1 with respect to rank $k$ is the truncated SVD of the weight error matrix:

$$\boldsymbol{C}_k = \boldsymbol{U}_{:,:k} \boldsymbol{\Sigma}_{:k,:k} \boldsymbol{V}_{:k,:}^T \tag{7}$$

where $\boldsymbol{U}$, $\boldsymbol{\Sigma}$, and $\boldsymbol{V}^T$ form the SVD of the weight quantization error, $\boldsymbol{U\Sigma V}^T = \text{SVD}(\boldsymbol{W} - \widetilde{\boldsymbol{W}})$.

As noted in Section 2, most existing works (Li et al., 2023; Yao et al., 2023; Guo et al., 2023) in QPEFT and PTQ adopt this solution. However, we know that minimizing the weight approximation error is not equivalent to minimizing the layer output error. Furthermore, does minimizing the weight approximation error for each layer in a network effectively reduce the final model output error? We will show that the answer is negative in Section 4.2.

**Problem 2** (Minimization of layer output error). For a pretrained linear layer $\boldsymbol{y} = \boldsymbol{xW}$ and its approximated form $\widetilde{\boldsymbol{y}} = \boldsymbol{x}(\widetilde{\boldsymbol{W}} + \boldsymbol{C}_k)$, approximating the layer by minimizing the error between $\boldsymbol{y}$ and $\widetilde{\boldsymbol{y}}$ is to minimize the following expectation.

$$\arg\min_{\boldsymbol{C}_k} \mathbb{E}_{\boldsymbol{y} \sim \mathbb{Y}} \{||\widetilde{\boldsymbol{y}} - \boldsymbol{y}||_2^2\} \tag{8}$$

where $|| \cdot ||_2$ denotes $l_2$ norm, and $\mathbb{Y} \subseteq \mathbb{R}^n$ is output space of the layer. We expand Equation (8) by substituting $\widetilde{\boldsymbol{y}}$ and $\boldsymbol{y}$:

$$\arg\min_{\boldsymbol{C}_k} \mathbb{E}_{\boldsymbol{x} \sim \mathbb{X}} \{||\boldsymbol{x}(\widetilde{\boldsymbol{W}} + \boldsymbol{C}_k) - \boldsymbol{xW}||_2^2\} \tag{9}$$

where $\mathbb{X} \subseteq \mathbb{R}^m$ is the input space. In practice, the expectation can be approximated as a sample mean on a calibration dataset like a subset of the pretraining data set.

Problem 2 motivates some recent works (Liu et al., 2023a; Guo et al., 2023; Zhang et al., 2024a) to involve activation-induced heuristics in the optimization of $\boldsymbol{C}_k$ *but without a theoretical foundation*. In the following two sections, we will derive the analytical solution to Problem 2. More precisely, we present two solutions: one exact solution in Section 3.2 and an approximated solution based on a suitable statistical assumption in Section 3.3.

## 3.2 QERA-EXACT: ANALYTICAL SOLUTION

QERA-exact is our exact solution to Problem 2. QERA-exact is computationally expensive as it calculates the autocorrelation matrix of the input space $\mathbb{X}$. However, as we will show in Section 4, QERA-exact recovers significant model performance in extremely low-precision quantization.

**Theorem 1** (QERA-exact solution). *The solution to Problem 2 is*

$$\boldsymbol{C}_k = \left(\boldsymbol{R}_{\mathbb{XX}}^{\frac{1}{2}}\right)^{-1} \boldsymbol{U}_{:,:k} \boldsymbol{\Sigma}_{:k,:k} \boldsymbol{V}_{:k,:}^T \tag{10}$$

where $\boldsymbol{R}_{\mathbb{XX}}$ is the autocorrelation matrix respect to the input space $\mathbb{X}$,

$$\boldsymbol{R}_{\mathbb{XX}} = \mathbb{E}_{\boldsymbol{x} \sim \mathbb{X}} \left\{ \boldsymbol{x}^T \boldsymbol{x} \right\} \tag{11}$$

$\boldsymbol{R}_{\mathbb{XX}}^{\frac{1}{2}}$ represents the unique symmetric positive semi-definite matrix square root of $\boldsymbol{R}_{\mathbb{XX}}$, and $\boldsymbol{U}_{:,:k}$, $\boldsymbol{\Sigma}_{:k,:k}$, and $\boldsymbol{V}_{:k,:}$ form the truncated SVD of the following scaled weight error matrix,

$$\boldsymbol{U}\boldsymbol{\Sigma}\boldsymbol{V}^T = \mathrm{SVD}(\boldsymbol{R}_{\mathbb{XX}}^{\frac{1}{2}}(\boldsymbol{W} - \widetilde{\boldsymbol{W}})) \tag{12}$$

**Remark 1.** $\boldsymbol{R}_{\mathbb{XX}}^{\frac{1}{2}}$ *is positive semi-definite. In the event that it has a zero eigenvalue, it would be normal to add a small diagonal perturbation to recover invertibility. In practice, we ran extensive experiments and find that $\boldsymbol{R}_{\mathbb{XX}}^{\frac{1}{2}}$ is invertible for all the pretrained models and datasets we present in Section 4.*

**Proof of Theorem 1**

*Proof.* Define $\boldsymbol{P} := \widetilde{\boldsymbol{W}} + \boldsymbol{C}_k - \boldsymbol{W}$, and $\boldsymbol{p}_i := \boldsymbol{P}_{i,:}$ is the $i$-th row of $\boldsymbol{P}$. Then we substitute $(\widetilde{\boldsymbol{W}} + \boldsymbol{C}_k - \boldsymbol{W})$ in the expanded objective Equation (9) of Problem 2 with $\boldsymbol{P}$:

$$\begin{aligned}
\mathbb{E}_{\boldsymbol{y} \sim \mathbb{Y}}\{||\widetilde{\boldsymbol{y}} - \boldsymbol{y}||_2^2\} &= \mathbb{E}_{\boldsymbol{x} \sim \mathbb{X}}\{||\boldsymbol{x}\boldsymbol{P}||_2^2\} \\
&= \mathbb{E}_{\boldsymbol{x} \sim \mathbb{X}}\{||\sum_{i=1}^m x_i \boldsymbol{p}_i||_2^2\} \\
&= \mathbb{E}_{\boldsymbol{x} \sim \mathbb{X}}\left\{ \sum_{i=1}^m \sum_{j=1}^m x_i x_j \boldsymbol{p}_i \boldsymbol{p}_j^T \right\}
\end{aligned} \tag{13}$$

We rewrite the last line of Equation (13) as:

$$\mathbb{E}_{\boldsymbol{y} \sim \mathbb{Y}}\{||\widetilde{\boldsymbol{y}} - \boldsymbol{y}||_2^2\} = \mathbb{E}_{\boldsymbol{x} \sim \mathbb{X}} \left\{ \boldsymbol{e} \cdot \left( (\boldsymbol{x}^T \boldsymbol{x}) \odot (\boldsymbol{P}\boldsymbol{P}^T) \right) \cdot \boldsymbol{e}^T \right\} \tag{14}$$

where $\boldsymbol{e} = \begin{bmatrix} 1 & 1 & \dots & 1 \end{bmatrix}$ is a row vector of $m$ ones, and $\odot$ denotes the element-wise product.

Using the property of the element-wise product (Styan, 1973), the RHS of the above can be simplified.

$$\begin{aligned}
\mathbb{E}_{\boldsymbol{y} \sim \mathbb{Y}}\{||\widetilde{\boldsymbol{y}} - \boldsymbol{y}||_2^2\} &= \mathbb{E}_{\boldsymbol{x} \sim \mathbb{X}} \left\{ \mathrm{Tr}\left( (\boldsymbol{x}^T \boldsymbol{x})(\boldsymbol{P}\boldsymbol{P}^T)^T \right) \right\} \\
&= \mathrm{Tr}\left( \mathbb{E}_{\boldsymbol{x} \sim \mathbb{X}} \left\{ \boldsymbol{x}^T \boldsymbol{x} \right\} \boldsymbol{P}\boldsymbol{P}^T \right) \\
&= \mathrm{Tr}\left( \boldsymbol{R}_{\mathbb{XX}} \boldsymbol{P}\boldsymbol{P}^T \right)
\end{aligned} \tag{15}$$

where $\mathrm{Tr}(\cdot)$ denotes trace and $\boldsymbol{R}_{\mathbb{XX}} = \mathbb{E}_{\boldsymbol{x} \sim \mathbb{X}} \left\{ \boldsymbol{x}^T \boldsymbol{x} \right\}$ is the autocorrelation matrix with respect to the input space $\mathbb{X}$.

Since $\boldsymbol{R}_{\mathbb{XX}}$ is a symmetric positive semi-definite matrix, it always has precisely one matrix square root, denoted as $\boldsymbol{R}_{\mathbb{XX}}^{\frac{1}{2}}$, that is also symmetric and positive semi-definite (Horn & Johnson, 2012). We reorganize Equation (15) as the following since both $\boldsymbol{R}_{\mathbb{XX}}$ and $(\boldsymbol{P}\boldsymbol{P}^T)$ are symmetric and positive semi-definite:

$$\begin{aligned}
\mathbb{E}_{\boldsymbol{y} \sim \mathbb{Y}}\{||\widetilde{\boldsymbol{y}} - \boldsymbol{y}||_2^2\} &= \mathrm{Tr}\left( \boldsymbol{R}_{\mathbb{XX}}^{\frac{1}{2}} \boldsymbol{P}\boldsymbol{P}^T \boldsymbol{R}_{\mathbb{XX}}^{\frac{1}{2}} \right) \\
&= \mathrm{Tr}\left( \boldsymbol{R}_{\mathbb{XX}}^{\frac{1}{2}} \boldsymbol{P}\boldsymbol{P}^T (\boldsymbol{R}_{\mathbb{XX}}^{\frac{1}{2}})^T \right) \\
&= ||\boldsymbol{R}_{\mathbb{XX}}^{\frac{1}{2}} \boldsymbol{P}||_F^2
\end{aligned} \tag{16}$$

Now the objective of Problem 2 ( Equation (8)) is equivalent to:

$$\begin{aligned}
\underset{\boldsymbol{C}_k}{\arg\min} \, \mathbb{E}_{\boldsymbol{y} \sim \mathbb{Y}}\{||\widetilde{\boldsymbol{y}} - \boldsymbol{y}||_2^2\} &= \underset{\boldsymbol{C}_k}{\arg\min} \, ||\boldsymbol{R}_{\mathbb{XX}}^{\frac{1}{2}} \boldsymbol{P}||_F^2 \\
&= \underset{\boldsymbol{C}_k}{\arg\min} \, ||\boldsymbol{R}_{\mathbb{XX}}^{\frac{1}{2}} (\widetilde{\boldsymbol{W}} + \boldsymbol{C}_k - \boldsymbol{W})||_F^2
\end{aligned} \tag{17}$$

If we assign $\boldsymbol{Q} := \boldsymbol{R}_{\mathbb{XX}}^{\frac{1}{2}}(\boldsymbol{W} - \widetilde{\boldsymbol{W}})$ and $\boldsymbol{Q}_k := \boldsymbol{R}_{\mathbb{XX}}^{\frac{1}{2}} \boldsymbol{C}_k$, the objective becomes:

$$\underset{\boldsymbol{Q}_k}{\arg\min} \, ||\boldsymbol{Q}_k - \boldsymbol{Q}||_F^2 \tag{18}$$

Note that multiplication by the invertible matrix $\boldsymbol{R}_{\mathbb{XX}}^{\frac{1}{2}}$ (Remark 1) does not change the rank of the matrix $\boldsymbol{C}_k$. According to the Eckart-Young-Mirsky theorem (Eckart & Young, 1936), the optimal rank $k$ approximation to $\boldsymbol{Q}_k$ is the truncated SVD of $\boldsymbol{Q}$:

$$\boldsymbol{Q}_k = \boldsymbol{U}_{:,:k} \boldsymbol{\Sigma}_{:k,:k} \boldsymbol{V}_{:k,:}^T \tag{19}$$

where $\boldsymbol{U}\boldsymbol{\Sigma}\boldsymbol{V}^T = \mathrm{SVD}(\boldsymbol{Q}) = \mathrm{SVD}\left(\boldsymbol{R}_{\mathbb{XX}}^{\frac{1}{2}}(\boldsymbol{W} - \widetilde{\boldsymbol{W}})\right)$. Thus the optimal rank-$k$ solution to $\boldsymbol{C}_k$ is:

$$\boldsymbol{C}_k = \left(\boldsymbol{R}_{\mathbb{XX}}^{\frac{1}{2}}\right)^{-1} \boldsymbol{Q}_k = \left(\boldsymbol{R}_{\mathbb{XX}}^{\frac{1}{2}}\right)^{-1} \boldsymbol{U}_{:,:k} \boldsymbol{\Sigma}_{:k,:k} \boldsymbol{V}_{:k,:}^T \tag{20}$$

$\square$

In practice, we assign $\boldsymbol{A}_k := \left(\boldsymbol{R}_{\mathbb{XX}}^{\frac{1}{2}}\right)^{-1} \boldsymbol{U}_{:,:k}$ and $\boldsymbol{B}_k := \boldsymbol{\Sigma}_{:k,:k} \boldsymbol{V}_{:k,:}^T$. Note that QERA adds no constraints to the quantization (and dequantization) function $\mathrm{q}(\cdot)$ (and $\mathrm{dq}(\cdot)$), *i.e.*, the low-precision $\widetilde{\boldsymbol{W}}$ can be obtained by any quantization method.

## 3.3 QERA-APPROX: AN ANALYTICAL SOLUTION WITH THE UNCORRELATED ASSUMPTION

QERA-approx is our analytical solution to Problem 2 based on the assumption that different embedding dimensions are uncorrelated. This solution is more computationally efficient than the exact solution, and the assumption is testable on real-world datasets. The complete proof of QERA-approx is in Appendix A.2.

**Assumption 1.** For a pretrained linear layer $\boldsymbol{y} = \boldsymbol{x}\boldsymbol{W}$, the expectation of the product of different embedding dimensions is zero:

$$\mathbb{E}_{\boldsymbol{x} \sim \mathbb{X}}\{x_i x_j\} = 0, \quad \forall i \neq j \tag{21}$$

where $x_i$ and $x_j$ are the $i$-th and $j$-th elements of the input vector $\boldsymbol{x}$.

We test this assumption on LLMs in Section 5.

**Theorem 2** (QERA-approx solution). *The solution to Problem 2 based on Assumption 1 is:*

$$\boldsymbol{C}_k = \boldsymbol{S}^{-1} \boldsymbol{U}_{:,:k} \boldsymbol{\Sigma}_{:k,:k} \boldsymbol{V}_{:k,:}^T \tag{22}$$

*where $\boldsymbol{S}$ is a diagonal matrix built from activation statistics,*

$$\boldsymbol{S} = \mathrm{diag}(\sqrt{\mathbb{E}_{\boldsymbol{x} \sim \mathbb{X}}\{x_1^2\}}, \sqrt{\mathbb{E}_{\boldsymbol{x} \sim \mathbb{X}}\{x_2^2\}}, \ldots, \sqrt{\mathbb{E}_{\boldsymbol{x} \sim \mathbb{X}}\{x_m^2\}}) \tag{23}$$

*and $\boldsymbol{U}$, $\boldsymbol{\Sigma}$, $\boldsymbol{V}^T$ form the SVD of the following scaled weight error matrix,*

$$\boldsymbol{U}\boldsymbol{\Sigma}\boldsymbol{V}^T = \mathrm{SVD}(\boldsymbol{S}(\boldsymbol{W} - \widetilde{\boldsymbol{W}})) \tag{24}$$

**Remark 2.** *For the diagonal matrix $\boldsymbol{S}$ in Theorem 2 to be invertible, we need $\mathbb{E}_{\boldsymbol{x} \sim \mathbb{X}}\{x_i^2\} \neq 0$ for all dimension $i$. In practice, this is almost always true for pretrained layers because no dimension in the input embeddings is always zero.*

For implementation, we assign $\boldsymbol{A}_k := \boldsymbol{S}^{-1} \boldsymbol{U}_{:,:k}$ and $\boldsymbol{B}_k := \boldsymbol{\Sigma}_{:k,:k} \boldsymbol{V}_{:k,:}$ to form the low-rank terms to save the memory and computation cost. Interestingly, QERA-approx solution is similar to the activation-induced heuristics in LQER (Zhang et al., 2024a), which calibrates the average absolute value on the embedding dimension (Refer to Algorithm 2 in the Appendix). In Section 4.3, we will show that our solution is more effective in practice and resolves the discrepancy between the recovered model performance and the number of calibration samples in LQER.

## 4 EXPERIMENTS

In this section, we first introduce the experiment setup in Section 4.1. Then we present the results of our experiments on QPEFT and PTQ in Section 4.2 and Section 4.3 respectively.

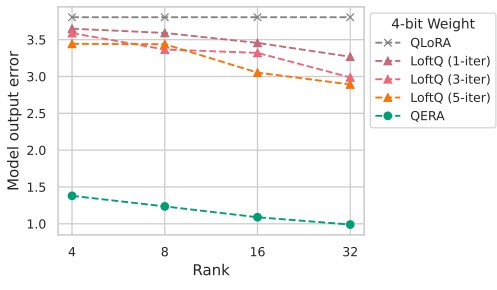 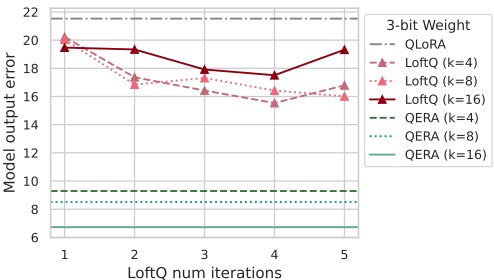

(a) Model output error *vs.* rank

(b) Model output error *vs.* LoftQ iterations

Figure 1: The model output error of RoBERTa-base before fine-tuning. We feed 128 samples from RoBERTa's pretraining dataset and profile the output logits error between the adapted and the FP32 model. We sweep the rank $k$ and the iteration number of LoftQ on 4-bit and 3-bit models. In LoftQ, neither more iterations nor a higher rank guarantees lower model output error, though the weight approximation error of every layer decreases. In contrast, QERA-approx consistently has the lowest model output error across all settings, and the error monotonically decreases as the rank increases.

## 4.1 EXPERIMENT SETUP

We perform QPEFT and PTQ experiments separately, and compare with their respective SoTA methods. The experiments take around 6400 GPU hours in total. The hardware platform, separate GPU hours, software dependencies, and random seed settings can be found in Appendix A.4.

For QPEFT experiments, we use Theorem 2, noted as QERA-approx, to initialize low-rank terms, and compare with full-finetuning, LoRA (Hu et al., 2021), QLoRA (Dettmers et al., 2024), and LoftQ (Li et al., 2023). Specifically, we adopt 5-iteration LoftQ, which is the officially recommended setup. We include both encoder-only model experiments (fine-tuning RoBERTa-base (Liu, 2019) on GLUE (Ye et al., 2019)) and decoder-only LLM experiments (fine-tuning LLaMA-2 (Touvron et al., 2023) and LLaMA-3.1 (Dubey et al., 2024) on continuous pretraining task SlimPajama (Soboleva et al., 2023) and supervised fine-tuning task GSM8K (Cobbe et al., 2021)). For each method/baseline, we sweep the learning rate and record the best result. The final results are averaged over three random seeds. The learning rate ranges and batch sizes are listed in Appendix A.4.1.

For PTQ experiments, we use both Theorem 1, noted as QERA-exact, and Theorem 2 (QERA-approx) to calculate the low-rank error reconstruction terms and report results separately. We compare with BF16, quantized model without error reconstruction terms ($w$-only), ZeroQuant-V2 (Yao et al., 2023), and LQER (Zhang et al., 2024a) at different precision setups. We also include HQQ (Badri & Shaji, 2023), a leading 4-bit method that does not use quantization error reconstruction. We quantize LLMs of various sizes and model family, including TinyLlama (Zhang et al., 2024b), Gemma-2 (Team et al., 2024), Phi-3.5 (Abdin et al., 2024) and LLaMA-2/-3.1 (Touvron et al., 2023; Dubey et al., 2024). We use `lm-evaluation-harness` to report results on Wikitext2 (Merity et al., 2016), ARC (challenge) (Clark et al., 2018), BoolQ (Clark et al., 2019), CommonSenseQA (Talmor et al., 2019), Winogrande (Sakaguchi et al., 2019), MMLU (Hendrycks et al., 2021), and BigBench-Hard (Suzgun et al., 2022). We also evaluate instruction-tuned model, Vicuna-v1.5 (Zheng et al., 2023), with AlpacaEval 2.0 (Dubois et al., 2024), which is an automatic evaluation tool for instruction-following tasks. Detailed setup is in Appendix A.4.2.

## 4.2 IMPROVED QPEFT

We first identify a pitfall in the commonly-used iterative Algorithm 1, that is, minimizing the weight approximation error for each layer does not necessarily minimize the model output error. Then we show that our QERA initialization enables a clear reduction in the model output error at the start of fine-tuning, leading to better fine-tuned accuracy/perplexity and faster convergence.

**Reduced layer weight error $\neq$ reduced model output error** We apply 4-bit and 3-bit QLoRA, LoftQ, and QERA-approx to RoBERTa-base and inspect the *model output error* on RoBERTa's pretraining dataset before fine-tuning at rank $k = 4, 8, 16, 32$. For LoftQ, we also sweep the number of iterations from 1 to 5. In Figure 1, we observe that

- For LoftQ, given a specific rank, increasing the optimization iterations does not guarantee a reduced model output error. Though all the layers' weight approximation errors monoton-

Table 1: Fine-tuning results of RoBERTa-base on GLUE. QERA-approx outperforms LoftQ across all bit widths, and the improvement is more obvious with aggressive quantization. QERA achieves $\Delta_{acc} = 4.12\%$ higher than LoftQ at 3-bit and 6.05% at 2-bit.

| Rank | W-bits | Method | MNLI Acc | QNLI Acc | RTE Acc | SST Acc | MRPC Acc | CoLA Matt | QQP Acc | STSB P/S Corr | Avg. |
|------|--------|--------|----------|----------|---------|---------|----------|-----------|---------|---------------|------|
| - | 16 | Full FT | 87.61 | 92.95 | 73.16 | 94.88 | 92.15 | 60.41 | 91.61 | 90.44/90.25 | 85.38 |
| 8 | 16 | LoRA | 87.85 | 92.84 | 69.55 | 94.46 | 89.99 | 57.52 | 89.83 | 89.92/89.83 | 84.00 |
| 8 | 4.25 | QLoRA | 87.21 | 92.32 | 63.90 | 94.08 | 88.24 | **56.08** | **90.55** | 89.59/89.56 | 82.75 |
|   |   | LoftQ (5-iter) | 87.27 | **92.48** | 67.13 | **94.38** | 88.24 | 54.59 | 90.51 | 88.75/88.79 | 82.92 |
|   |   | QERA-approx | **87.28** | 92.45 | **70.40** | **94.38** | **88.97** | 55.99 | 90.39 | **89.83/89.72** | **83.71** |
| 8 | 3.25 | QLoRA | 84.87 | 89.58 | 53.67 | 91.02 | 73.94 | 3.12 | 89.31 | 84.80/84.38 | 71.29 |
|   |   | LoftQ (5-iter) | 85.24 | 89.65 | 58.24 | 92.05 | 75.82 | 11.00 | 88.93 | 85.55/85.27 | 73.31 |
|   |   | QERA-approx | **85.58** | **90.74** | **58.48** | **92.59** | **82.19** | **32.98** | **89.41** | **87.43/87.08** | **77.43** |
| 64 | 2.50 | QLoRA | 77.87 | 85.26 | 54.15 | 90.02 | 71.00 | 0 | 87.93 | 74.72/75.31 | 67.62 |
|   |   | LoftQ (5-iter) | 80.15 | 87.65 | 52.95 | 90.94 | 74.35 | 3.43 | 89.17 | 82.76/82.90 | 70.18 |
|   |   | QERA-exact | **84.64** | **90.05** | **58.48** | **92.32** | **84.72** | **26.43** | **89.69** | **86.48/86.40** | **76.23** |

Table 2: Fine-tuning results of LLaMA-2-7B and LLaMA-3.1-8B on SlimPajama and GSM8K. A trend similar to RoBERTa experiments are observed, *i.e.*, QERA outperforms QLoRA and LoftQ and the improvement is more obvious on aggressive quantization.

| W-bits | Method | LLaMA-2-7B | | LLaMA-3.1-8B | |
|--------|--------|------------|---|--------------|---|
| | | SlimPajama ($\Delta_{ppl}$) | GSM8K ($\Delta_{acc}$) | SlimPajama ($\Delta_{ppl}$) | GSM8K ($\Delta_{acc}$) |
| 16 | LoRA | 6.17 | 39.40 | 8.07 | 55.72 |
| 4.25 | QLoRA | 6.44 (+0.27) | 30.71 (-8.69) | 8.70 (+0.63) | 54.81 (-0.91) |
| | LoftQ (5-iter) | 6.39 (+0.22) | 28.58 (-10.82) | 8.73 (+0.66) | 54.23 (-1.49) |
| | QERA-approx | **6.33 (+0.16)** | **32.26 (-7.14)** | **8.68 (+0.61)** | **55.24 (-0.48)** |
| 2.25 | QLoRA | 53.95 (+47.78) | 12.79 (-18.31) | 71.90 (+63.83) | 5.08 (-50.64) |
| | LoftQ (5-iter) | 12.30 (+6.13) | 18.37 (-12.73) | 27.16 (+19.09) | 13.72 (-42.00) |
| | QERA-approx | **10.56 (+4.39)** | 18.78 (-12.32) | **20.07 (+12.00)** | **19.41 (-36.31)** |

ically decrease with the number of iterations (as illustrated in Figure 6 in Appendix), the model output error does not monotonically decrease. For example, in Figure 1a, the model output error of LoftQ (5-iter) is larger than LoftQ (3-iter) at rank $k = 8$.

- For LoftQ, given a specific number of iterations, increasing the rank does not guarantee a reduced model output error. For example, in Figure 1b, the output error of LoftQ (rank $k = 16$) is larger than LoftQ (rank $k = 4$) and $k = 8$ at 2, 3, 4, and 5 iterations.

- The model output error of our QERA-approx is always smaller than LoftQ and QLoRA, across all precision and rank settings. Moreover, the output error of QERA-approx monotonically decreases as the rank increases.

This empirical evidence suggests a strong correlation between the reduction of layer output error and the decrease in model output error in QER problem. Conversely, minimizing weight approximation error using LoftQ does not have a comparable impact on overall model performance.

**Better optimization quality** Table 1 and Table 2 summarize the fine-tuning experiments of RoBERTa-base on GLUE, and LLaMA-2-7B/-3.1-8B fine-tuned on SlimPajama and GSM8K, respectively. QERA outperforms both LoftQ and QLoRA. In GLUE experiments, at 4-bit, QERA enables an average accuracy gain of 0.96% and 0.79% higher than QLoRA and LoftQ respectively, close to BF16 LoRA; At 3-bit and 2-bit, QERA achieves a 4.12% and 6.05% higher average accuracy than LoftQ respectively. Similar trends are observed on LLM fine-tuning experiments, *i.e.*, QERA outperforms QLoRA and LoftQ, and the advantage of QERA over LoftQ is more obvious with more aggressive quantization.

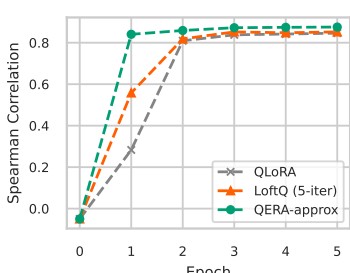

Figure 2: Faster convergence of QERA-approx on STSB.

**Faster Convergence** QERA initialization also speeds up the training convergence. For LLM fine-tuning, this is expected as QERA initialization is closer to the full-precision model. Interestingly, in encoder-only experiments on GLUE where the model

Table 3: Perplexity (↓) of LLMs on WikiText2. $w$-only denotes the quantized model without low-rank error reconstruction. QERA-approx outperforms LQER on almost all setups and QERA-exact achieves the lowest perplexity. The advantage of QERA is pronounced at 3-bit.

| W-bits | Method | Rank | TinyLlama | Gemma-2 | Phi-3.5 | LLaMA-2 | | LLaMA-3.1 | |
|---|---|---|---|---|---|---|---|---|---|
| | | | 1.1B | 2B | 3.8B | 7B | 13B | 8B | 80B |
| - | BF16 | - | 13.98 | 13.08 | 11.50 | 8.71 | 7.68 | 7.55 | 3.06 |
| 4.25 | HQQ | - | *15.02* | 14.29 | 14.63 | 9.59 | 8.27 | 8.72 | 3.97 |
| 4.25 | $w$-only | - | 19.40 | 16.23 | 14.16 | 9.45 | 8.06 | 8.78 | 4.55 |
| | ZeroQuant-V2 | | 18.03 | 15.71 | 14.09 | 9.42 | 8.07 | 8.83 | 4.48 |
| | LQER | 32 | 16.23 | 14.55 | 12.88 | 9.22 | 7.96 | 8.45 | 4.10 |
| | QERA-approx | | **15.66** | 14.60 | 12.81 | 9.17 | 7.95 | 8.45 | 4.10 |
| | QERA-exact | | 16.16 | *14.12* | *12.30* | *9.12* | *7.93* | *8.33* | *3.82* |
| 3.25 | $w$-only | - | 32.82 | 41.13 | 47.78 | 13.32 | 10.24 | 18.96 | 16.46 |
| | ZeroQuant-V2 | | 27.80 | 33.56 | 42.64 | 13.00 | 10.03 | 19.29 | 10.12 |
| | LQER | 64 | 20.60 | 21.99 | 18.27 | 14.00 | 9.09 | 11.86 | 7.05 |
| | QERA-approx | | 20.43 | 21.93 | **17.99** | 10.99 | 9.04 | 11.73 | 6.99 |
| | QERA-exact | | **19.51** | **19.97** | 20.37 | **10.67** | **8.97** | **11.39** | **6.68** |

classifier head is randomly initialized, we also observe that QERA converges faster, especially on small subsets such as STSB and MRPC where only a few thousand samples are available (in comparison MNLI has 393k samples and QQP has 364k samples). For example, in Figure 2, the Spearman correlation coefficient of QERA on STSB increases and converges faster than LoftQ and QLoRA, as the green line plateaus first.

## 4.3 IMPROVED PTQ

In this part, we first demonstrate that LQER, which depends on heuristics derived from activation values, does not guarantee improved performance with a larger calibration dataset. However, QERA exhibits the opposite trend. Through extensive experiments, we show that QERA consistently outperforms ZeroQuant-V2 and LQER, and QERA-exact exhibits better model performance than QERA-approx at the cost of more computation in the quantization process. These results verified the effectiveness of our analytical solution.

**Model performance *vs.* calibration set size** As mentioned at the end of Section 3.3, the scale matrix in LQER (Zhang et al., 2024a) is similar to the one in QERA-approx, but is based on hand-crafted heuristics. As a result, we observe that the model performance of LQER varies randomly as the number of calibration samples increases (the purple curve in Figure 3). On the contrary, more calibration samples consistently lead to better model performance for QERA until convergence.

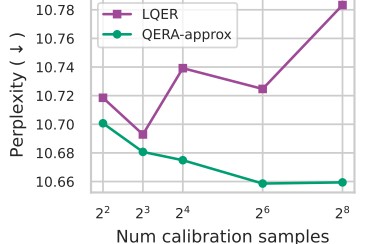

Figure 3: QERA resolves the discrepancy between the recovered model performance and the number of calibration samples in LQER.

**Improved perplexity and downsteam task accuracy** We apply QERA-approx and QERA-exact to a range of models and evaluate on both pretraining task and downstream tasks in Table 3 and Table 4 respectively. We also compare to HQQ, a SoTA method that does not use quantization error reconstruction. On most models, QERA-approx outperforms ZeroQuant-V2 and LQER, while QERA-exact achieves the best performance. At 4-bit, QERA-exact is nearly lossless. At 3-bit, QERA-exact's improvement over QERA-approx (Table 3) is clear, indicating the superiority of QERA-exact for aggressive quantization.

**Higher win rate on AlpacaEval 2.0** To better understand the impact on instruction-tuned models, we present the results of Vicuna-7b-v1.5 on AlpacaEval 2.0. In Figure 4, we evaluate the QER-based methods against the $w$-only quantization counterpart. QERA outperforms ZeroQuant-V2 and LQER by a higher win rate, indicating a better response quality.

---

[1]The average accuracy of TinyLlama-1.1B excludes BoolQ, CommonsenseQA, and MMLU since TinyLlama-1.1B has random guess accuracy on these tasks.

Table 4: Average accuracy (↑) of LLMs on six downstream tasks. QERA-exact outperforms other quantization-error reconstruction-based methods across almost all models. We also compare to HQQ (Badri & Shaji, 2023), a SoTA PTQ method that does not adopt quantization-error reconstruction or activation heuristics. QERA-exact achieves an average accuracy on par with HQQ.

| W-bits | Method | Rank | TinyLlama[1] | Gemma-2 | Phi-3.5 | LLaMA-2 | | LLaMA-3.1 | |
|--------|--------|------|-----------|---------|---------|---------|------|-----------|------|
| | | | 1.1B | 2B | 3.8B | 7B | 13B | 8B | 80B |
| - | BF16 | - | 40.59 | 53.96 | 66.91 | 49.61 | 55.74 | 63.88 | 72.05 |
| 4.25 | HQQ | - | 40.35 | *52.54* | 59.17 | 48.26 | 54.53 | *62.59* | 71.31 |
| 4.25 | *w*-only | - | 36.56 | 48.33 | 64.52 | 47.62 | 55.12 | 61.53 | 68.46 |
| | ZeroQuant-V2 | | 37.26 | 48.24 | 64.44 | 47.43 | 55.15 | 61.70 | 68.45 |
| | LQER | 32 | ***40.45*** | 49.77 | 64.46 | 48.47 | 55.40 | 61.75 | 70.94 |
| | QERA-approx | | 40.02 | 49.29 | 64.53 | 48.52 | 55.20 | 61.68 | 70.80 |
| | QERA-exact | | 40.36 | **51.73** | **65.08** | ***48.91*** | ***55.42*** | **62.05** | ***71.42*** |

## 5  DISCUSSION

In this section, we revisit the arguments, design choices, and observations made in the previous sections, including a test of Assumption 1, and the choice of the calibration set for PEFT. We offer an extended discussion of the numeric stability and scalability in Appendix A.7, and LoRA rank and model choices of PEFT experiments in Appendix A.9.

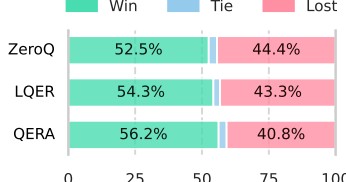

**Test of Assumption 1**  To test Assumption 1, we profile the autocorrelation matrix $\boldsymbol{R}_{\mathbb{XX}}$ of the linear layer inputs in LLaMA-2-7B and LLaMA-3-8B. Note that $\mathbb{R}_{\mathbb{XX}i,j} = \mathbb{E}_{\boldsymbol{x} \sim \mathbb{X}}\{x_i x_j\}$, which assumes to be zero for $i \neq j$ in Assumption 1. Figure 5 shows the normalized $\mathbb{R}_{\mathbb{XX}}$ magnitude, $\frac{\text{abs}(\mathbb{R}_{\mathbb{XX}})}{\|\mathbb{R}_{\mathbb{XX}}\|_F}$, of four representative layers in LLaMA-3-8B where darker elements denote values closer to zero.

Figure 4: AlpacaEval 2.0 evaluation results. We compare quantized models to the counterpart without quantization-error reconstruction. A higher win rate (↑) indicates better instruction-following performance.

There are several layers with some input dimensions strongly correlated with others, such as the inputs to the third attention layer in Figure 5a, but for most layers, our assumption holds, especially the MLP layers, such as Figures 5b to 5d. More $\mathbb{R}_{\mathbb{XX}}$ plots are in Appendix A.11.

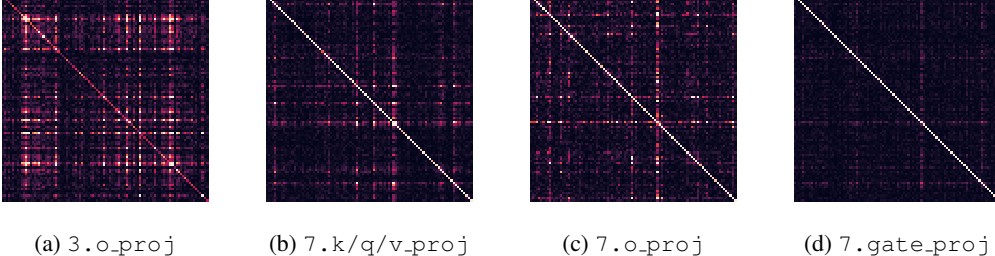

| (a) 3.o_proj | (b) 7.k/q/v_proj | (c) 7.o_proj | (d) 7.gate_proj |

Figure 5: Normalized $\text{abs}(\boldsymbol{R}_{\mathbb{XX}})$ of the layer inputs in LLaMA-3-8B. Dark elements denotes value close to zero. There are a few layers with input dimensions strongly correlated with others, such as the third attention layer in (a), but for most layers, our assumption of zero-expectation holds.

**Choice of calibration set for QPEFT**  One problem is to determine the calibration set for QERA before fine-tuning. In 2-bit RoBERTa-base fine-tuning experiment on SST2 (Appendix A.6), we find that calibrating on the pretraining dataset, WikiText2, helps the loss to decrease. However, the loss of the model calibrated on the fine-tuning dataset does not follow the same trend. We hypothesize that the massive padding tokens in preprocessed SST2 samples cause this discrepancy, especially considering that the sequence length of the raw SST2 dataset changes fiercely.

## 6  CONCLUSION

In this paper, we formulate the problem of quantization error reconstruction and propose QERA as an analytical solution. Applying QERA to related works for efficient fine-tuning or inference, we show that QERA resolves the discrepancy in existing methods, and outperforms SoTA methods in both fine-tuning and quantization tasks by a clear margin.

ACKNOWLEDGMENTS

This work was sponsored by Advanced Research + Invention Agency (ARIA), UK. We also thank ARIA for their research network.

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

# A APPENDIX

## A.1 ALGORITHMS IN RELATED WORK

Here we summarize the algorithm of LoftQ (Li et al., 2023) in Algorithm 1 and LQER (Zhang et al., 2024a) in Algorithm 2 respectively. LQ-LoRA (Guo et al., 2023) adopts a variant of Algorithm 1. ZeroQuant-V2 (Yao et al., 2023) can be considered as Algorithm 1 with one iteration, or a special case of Algorithm 2 where the scale matrix $\boldsymbol{S}$ is an identity matrix.

---

**Algorithm 1** LoftQ (Li et al., 2023)

---

**Require:** Pretrained weight $\boldsymbol{W}$, target rank $k$, quantization function $\mathrm{q}(\cdot)$, dequantization function $\mathrm{dq}(\cdot)$, number of iterations $T$
1: $\boldsymbol{A}_k \leftarrow \boldsymbol{0}, \boldsymbol{B}_k \leftarrow \boldsymbol{0}$
2: **for** $i = 1$ to $T$ **do**
3:      $\boldsymbol{W}_q \leftarrow \mathrm{q}(\boldsymbol{W} - \boldsymbol{A}_k \boldsymbol{B}_k)$              ▷ Update quantized weight matrix
4:      $\widetilde{\boldsymbol{W}} \leftarrow \mathrm{dq}(\boldsymbol{W}_q)$
5:      $\boldsymbol{U}, \boldsymbol{\Sigma}, \boldsymbol{V}^T \leftarrow \mathrm{SVD}(\boldsymbol{W} - \widetilde{\boldsymbol{W}})$              ▷ SVD-based rank-$k$ approximation
6:      $\boldsymbol{A}_k \leftarrow \boldsymbol{U}_{:,:k} \sqrt{\boldsymbol{\Sigma}_{:k,:k}}, \boldsymbol{B}_k \leftarrow \sqrt{\boldsymbol{\Sigma}_{:k,:k}} \boldsymbol{V}_{:k,:}^T$
7: **end for**

---

---

**Algorithm 2** LQER (Zhang et al., 2024a)

---

**Require:** Pretrained weight $\boldsymbol{W}$, target rank $k$, quantization function $\mathrm{q}(\cdot)$, dequantization function $\mathrm{dq}(\cdot)$, calibration dataset $\mathbb{X} = \{\boldsymbol{x}_i \in \mathbb{R}^m | i = 1, \ldots, N\}$
1: Initialize vector $\boldsymbol{s} \leftarrow \boldsymbol{0}$
2: **for** sample $\boldsymbol{x}$ in $\mathbb{X}$ **do**                                        ▷ Calibration
3:      $\boldsymbol{s} \leftarrow \boldsymbol{s} + \mathrm{abs}(\boldsymbol{x})$             ▷ Accumulate activation magnitude on each dimension
4: **end for**
5: $\boldsymbol{S} \leftarrow \frac{1}{N} \mathrm{diag}(\boldsymbol{s})$                                ▷ Construct a diagonal matrix $\boldsymbol{S}$
6: $\boldsymbol{W}_q \leftarrow \mathrm{q}(\boldsymbol{W})$
7: $\widetilde{\boldsymbol{W}} \leftarrow \mathrm{dq}(\boldsymbol{W}_q)$
8: $\boldsymbol{U}, \boldsymbol{\Sigma}, \boldsymbol{V}^T \leftarrow \mathrm{SVD}(\boldsymbol{S}(\boldsymbol{W} - \widetilde{\boldsymbol{W}}))$             ▷ SVD on the scaled weight error
9: $\boldsymbol{A}_k \leftarrow \boldsymbol{S}^{-1} \boldsymbol{U}_{:,:k}, \boldsymbol{B}_k \leftarrow \boldsymbol{\Sigma}_{:k,:k} \boldsymbol{V}_{:k,:}^T$            ▷ Rank-$k$ approximation with un-scaling

---

## A.2 PROOF OF THEOREM 2

Here we present the full proof of QERA-approx. QERA-approx is an approximated solution to Problem 2 based on Assumption 1, which is suitable to initialize the low-rank terms in fine-tuning for lower computation complexity.

**Proof of Theorem 2**

*Proof.* We continue at Equation (13). Since $\mathbb{E}_{\boldsymbol{x} \sim \mathbb{X}}$ is the expectation with respect to the input space, we move the expectation inside the summation of RHS of Equation (13).

$$\mathbb{E}_{\boldsymbol{y} \sim \mathbb{Y}}\{||\widetilde{\boldsymbol{y}} - \boldsymbol{y}||_2^2\} = \sum_{i=1}^{m} \sum_{j=1}^{m} \mathbb{E}_{\boldsymbol{x} \sim \mathbb{X}}\{x_i x_j\} \boldsymbol{p}_i \boldsymbol{p}_j^T \tag{25}$$

Under Assumption 1, $\mathbb{E}_{\boldsymbol{x} \sim \mathbb{X}}\{x_i x_j\} = 0$ for $i \neq j$, the RHS of Equation (25) simplifies to:

$$\mathbb{E}_{\boldsymbol{y} \sim \mathbb{Y}}\{||\widetilde{\boldsymbol{y}} - \boldsymbol{y}||_2^2\} = \sum_{i=1}^{m} \mathbb{E}_{\boldsymbol{x} \sim \mathbb{X}}\{x_i^2\} \boldsymbol{p}_i \boldsymbol{p}_i^T \tag{26}$$

We can define diagonal matrix $\boldsymbol{S} = \mathrm{diag}(\sqrt{\mathbb{E}_{\boldsymbol{x} \sim \mathbb{X}}\{x_1^2\}}, \sqrt{\mathbb{E}_{\boldsymbol{x} \sim \mathbb{X}}\{x_2^2\}}, \ldots, \sqrt{\mathbb{E}_{\boldsymbol{x} \sim \mathbb{X}}\{x_m^2\}})$ and rewrite the RHS of Equation (26) as:

$$\mathbb{E}_{\boldsymbol{y} \sim \mathbb{Y}}\{||\widetilde{\boldsymbol{y}} - \boldsymbol{y}||_2^2\} = \mathrm{Tr}(\boldsymbol{S} \boldsymbol{P} \boldsymbol{P}^T \boldsymbol{S}^T) = ||\boldsymbol{S} \boldsymbol{P}||_F^2 \tag{27}$$

where $\mathrm{Tr}(\cdot)$ denotes the trace of a matrix.

Therefore, the objective of Problem 2 (Equation (8)) is equivalent to:

$$\arg\min_{\boldsymbol{C}_k} \mathbb{E}_{\boldsymbol{y} \sim \mathbb{Y}}\{||\widetilde{\boldsymbol{y}} - \boldsymbol{y}||_2^2\} = \arg\min_{\boldsymbol{C}_k} ||\boldsymbol{SP}||_F^2$$
$$= \arg\min_{\boldsymbol{C}_k} ||\boldsymbol{S}(\widetilde{\boldsymbol{W}} + \boldsymbol{C}_k - \boldsymbol{W})||_F^2 \quad (28)$$

If we assign $\boldsymbol{Q} = \boldsymbol{S}(\boldsymbol{W} - \widetilde{\boldsymbol{W}})$ and $\boldsymbol{Q}_k = \boldsymbol{SC}_k$, the objective becomes:

$$\arg\min_{\boldsymbol{Q}} ||\boldsymbol{Q}_k - \boldsymbol{Q}||_F^2 \quad (29)$$

Note that the invertible matrix $\boldsymbol{S}$ in $\boldsymbol{Q}_k$ does not change the rank of the matrix $\boldsymbol{C}_k$. According to the Eckart-Young-Mirsky theorem, the optimal rank $k$ approximation to $\boldsymbol{Q}$ is the truncated SVD of $\boldsymbol{Q}$:

$$\boldsymbol{Q}_k = \boldsymbol{U}_{:,:k}\boldsymbol{\Sigma}_{:k,:k}\boldsymbol{V}_{:k,:}^T \quad (30)$$

where $\boldsymbol{U}\boldsymbol{\Sigma}\boldsymbol{V}^T = \text{SVD}(\boldsymbol{Q}) = \text{SVD}\left(\boldsymbol{S}(\boldsymbol{W} - \widetilde{\boldsymbol{W}})\right)$.

Finally, we get the optimal solution to the low-rank term $\boldsymbol{C}_k$:

$$\boldsymbol{C}_k = \boldsymbol{S}^{-1}\boldsymbol{Q}_k = \boldsymbol{S}^{-1}\boldsymbol{U}_{:,:k}\boldsymbol{\Sigma}_{:k,:k}\boldsymbol{V}_{:k,:}^T \quad (31)$$

$\square$

## A.3 Connection and Difference between CALDERA and QERA

CALDERA (Saha et al., 2024) is the concurrent work close to QERA. Here we elaborate the connection and difference between CALDERA and QERA, and highlight the contributions of QERA.

CALDERA focuses on a different problem setup. Specifically, CALDERA focuses on the following problem:

$$\min_{\widetilde{\boldsymbol{W}}, \boldsymbol{A}_{k,q}, \boldsymbol{B}_{k,q}} ||\boldsymbol{XW} - \boldsymbol{X}(\widetilde{\boldsymbol{W}} + \boldsymbol{A}_{q,k}\boldsymbol{B}_{q,k})||_F^2 \quad (32)$$

where $\boldsymbol{X} \in \mathbb{R}^{b \times m}$ denotes a batch of calibration samples, and $\widetilde{\boldsymbol{W}}$, $\boldsymbol{A}_{k,q}$, and $\boldsymbol{B}_{k,q}$ are all quantized variables to optimize. Note that this problem setup is different from QERA (Equation (9)):

$$\arg\min_{\boldsymbol{C}_k} \mathbb{E}_{\boldsymbol{x} \sim \mathbb{X}}\{||\boldsymbol{x}(\widetilde{\boldsymbol{W}} + \boldsymbol{C}_k) - \boldsymbol{x}\boldsymbol{W}||_2^2\} \quad (33)$$

where only the low-rank high-precision $\boldsymbol{C}_k := \boldsymbol{A}_k\boldsymbol{B}_k$ is the variable to optimize, and the quantized weight $\widetilde{\boldsymbol{W}}$ is predefined given a quantization method.

Table 5: Notation Table for the Equivalence Derivation

| Notation | Description | Comments |
|---|---|---|
| $b$ | Number of calibration samples (vectors) | |
| $m$ | Layer input feature size | |
| $n$ | Layer output feature size | |
| $\boldsymbol{X}$ | Calibration set | Shape: $b \times m$ |
| $\boldsymbol{x}$ | A sample in the calibration set | Shape: $1 \times m$ |
| $\boldsymbol{W}$ | Original full-precision layer weights | Shape: $m \times n$ |
| $\boldsymbol{Y}$ | Layer output matrix corresponding to $\boldsymbol{X}$ | Shape: $b \times n$ |
| $\boldsymbol{y}$ | Layer output vector corresponding to $\boldsymbol{x}$ | Shape: $1 \times n$ |
| $k$ | Rank of the low-rank approximation | |
| $\boldsymbol{C}_k$ | Approximated rank-$k$ weight | Shape: $m \times n$ |
| $\boldsymbol{U}, \boldsymbol{\Sigma}, \boldsymbol{V}$ | SVD decomposition of $\boldsymbol{X}$ | |
| $\text{SVD}_k(\cdot)$ | Truncated rank-$k$ SVD | |

We find that CALDERA's Lemma 4.2 is equivalent to Theorem 1 in QERA. Note that the proof of QERA-exact is different from Caldera's Lemma 4.2, though the final closed-form solution is

equivalent. Here we additionally show the derivation of the equivalence between QERA-exact and Caldera's Lemma 4.2 using the notation table in Table 5. For convenience, we remove the quantized weight term $\widetilde{W}$ from QERA (Problem 2 in Equation (9)), which does not change the proof. Now the problem becomes finding the optimal low-rank approximation of the weight matrix, $C_k$ that minimizes the layer output error.

First we note that the objective of QERA, Equation (9), is equivalent to CALDERA's Eq(5) scaled by a constant $n$:

$$
\begin{aligned}
\text{QERA} : \min_{C_k} E_{\mathbf{x}}\{||\mathbf{x}(C_k - W)||_2^2\} \\
\text{CALDERA} : \min_{C_k}||X(C_k - W)||_F^2
\end{aligned}
\tag{34}
$$

Then we show that Theorem 1 (QERA-exact) is equal to Caldera's Lemma 4.2.

$$
\text{QERA-exact} : C_k = (R_{\mathbb{XX}}^{\frac{1}{2}})^{-1} \cdot \text{SVD}_k(R_{\mathbb{XX}}^{\frac{1}{2}} W)
\tag{35}
$$

$$
\text{CALDERA} : C_k' = V\Sigma \cdot \text{SVD}_k(U^T Y)
\tag{36}
$$

We firstly show that $(R_{\mathbb{XX}}^{\frac{1}{2}})^{-1}$ in Equation (35) equals to $V\Sigma$ in Equation (36) scaled by a constant $\sqrt{b}$:

$$
\begin{aligned}
R_{\mathbb{XX}} &= \frac{1}{b}(X^T X) = V\Sigma U^T U\Sigma V^T = V\Sigma^2 V^T \\
R_{\mathbb{XX}}^{\frac{1}{2}} &= \frac{1}{\sqrt{b}}\Sigma V^T \\
(R_{\mathbb{XX}}^{\frac{1}{2}})^{-1} &= \sqrt{b}V\Sigma^{-1}
\end{aligned}
\tag{37}
$$

Then we show that $R_{\mathbb{XX}}^{\frac{1}{2}} W$ in Equation (35) equals to $U^T Y$ in Equation (36) scaled by the constant $\frac{1}{\sqrt{b}}$:

$$
\begin{aligned}
U^T Y &= U^T X W = U^T U\Sigma V^T W = \Sigma V^T W = \sqrt{b}R_{\mathbb{XX}}^{\frac{1}{2}} W \\
R_{\mathbb{XX}}^{\frac{1}{2}} W &= \frac{1}{\sqrt{b}}U^T Y
\end{aligned}
\tag{38}
$$

Therefore $C_k$ equals to $C_k'$, and the two solutions are equivalent. Despite of the equivalence, we shortlist the differences between CALDERA and our work:

- Different problem setup (Equation (34)).

- We simplify QERA-exact and derive QERA-approx, which is a computationally-efficient approximated solution. Specifically, QERA-approx is more suitable for parameter-efficient fine-tuning than QERA-exact/CALDERA. Moreover, QERA-approx overcomes the pitfalls in existing methods and explains why previous heuristic methods like LQER work.

- The optimization objective is similar (vector form *vs* matrix form), and the final closed-form solution is equivalent, but the proof of QERA-exact is different from CALDERA.

## A.4   DETAILED EXPERIMENT SETUP

We mainly use `PyTorch`, `Transformers`, `PEFT`, and `Accelerate` to implement QERA. We use `SciPy`'s implementation of blocked Schur algorithm (Deadman et al., 2012) to calculate the matrix square root, which runs on CPUs. The evaluation is performed with `lm-evaluation-harness`, `Evaluate`, and AlpacaEval 2.0 (Dubois et al., 2024).

### A.4.1   QPEFT HYPERPARAMETERS

We perform fine-tuning experiments on four NVIDIA A100 80GB GPUs with AMD EPYC 64-Core Processor with 1024GB RAM. The total fine-tuning time is around 2100 GPU hours.

**RoBERTa-base on GLUE**    We sweep learning rates for each (`method`, `task`), and collect the best accuracy. Thus each (`method`, `task`) pair has its own tailored learning rate, ensuring the best performance of baselines and QERA under the same trainable parameter budget. The reported accuracy is the average value across random seeds 42, 1, and 2. The total batch size is 64 for all GLUE experiments and we train the models for 5 epochs. For 4-bit experiments, we use 4-bit floating point from the QLoRA implementation in `PEFT`. For 3-bit experiments, we use emulated MXINT (Darvish Rouhani et al., 2023) with block size = 32 and for 2-bit experiments we use MXINT with block size = 16. Table 6 lists the learning rates for each experiment.

**LLaMA-2-7B/-3.1-8B on SlimPajama and GSM8K**    We adopt the learning rates in Meng et al. (2024). The reported perplexity/accuracy is the average value across random seeds 42, 1, and 2. For SlimPajama, we fine-tune the model on a subset for 1000 steps with rank = 8, total batch size = 64, sequence length = 1024, learning rate = 3e-5. For GSM8K, we fine-tune the model for 10 epochs with rank = 64, total batch size = 128, sequence length = 384, and learning rate = 3e-5.

Table 6: Learning rates of RoBERTa-base experiments on GLUE.

| Rank | W-bits | Method | Learning rates |
|---|---|---|---|
| - | 16 | Full FT | 7e-5, 5e-5, 2e-5 |
| 8 | 16 | LoRA | 1e-4, 2e-4, 3e-4 |
| 8 | 4.25 | QLoRA/LoftQ/QERA-approx | 1e-4, 2e-4, 3e-4 |
| 8 | 3.25 | QLoRA/LoftQ/QERA-approx | 1e-4, 2e-4, 3e-4 |
| 64 | 2.50 | QLoRA/LoftQ/QERA-exact | 2e-5, 3e-5, 4e-5, 5e-5, 6e-5, 9e-5, 1e-4 |

### A.4.2    PTQ Hyperparameters

We perform PTQ experiments on eight NVIDIA A6000 48GB GPUs with AMD EPYC 256-Core Processor with 1024GB RAM. The total quantization and evaluation time is around 4500 GPU hours. We report 0-shot accuracy or normalized accuracy (if available) for all tasks except Wiki-Text2, in which we report word perplexity. The sequence length for reporting word perplexity is the model's context length by default, except for Phi-3.5 and LLaMA-3.1. For these two models, we set the sequence length = 2048. We use the HuggingFace `Transformers`'s implementation of HQQ, and reimplement ZeroQuant-V2 and LQER as baselines. We use MXINT with block size = 32 as the quantization format for all quantization methods except HQQ, which uses its built-in INT format with group size = 64. Thus, both formats have an average W-bits of 4.25. We evaluate quantized Vicuna-v1.5-7B, which is an instruction-tuned LLaMA-2-7B, with AlpacaEval 2.0. and use GPT4-Turbo as the evaluator. The reported win rate is the length-controlled win rate, which is a debiased version of the win rate that controls for the length of the generated outputs.

### A.5    Decreasing Weight Error $\neq$ Decreasing Output Error for LoftQ

We provide the weight approximation error, $||\boldsymbol{W} - \widetilde{\boldsymbol{W}} - \boldsymbol{C}_k||_F$, in Figure 6, of all linaer layers in RoBERTa-base by sweeping the number of iterations. We observe that the weight approximation error monotonically decreases with the number of iterations, but as shown in Figure 1, the model output error may increase. This observation indicates that the commonly used objective of minimizing the weight approximation error and the corresponding algorithm are not ideal for the quantization error reconstruction problem.

### A.6    Choice of Calibration Set

We compare the QERA-adapted models calibrated on the pretraining dataset and the downstream dataset. Specifically, we fine-tune two QERA-adapted 2-bit RoBERTa-base models. One is calibrated on its pretraining dataset, WikiText2, and the other on SST2. Figure 7 shows the loss curves of the two models across three learning rates. None loss curves of the models calibrated on SST2 decreases, but the ones calibrated on WikiText2 successfully decrease and converge. We hypothesize that this is due to the massive padding tokens in preprocessed SST2 considering that the raw sample lengths change fiercely. However, WikiText2 samples were preprocessed in the masked language modeling style, which means that only a few special tokens are added to the grouped texts.

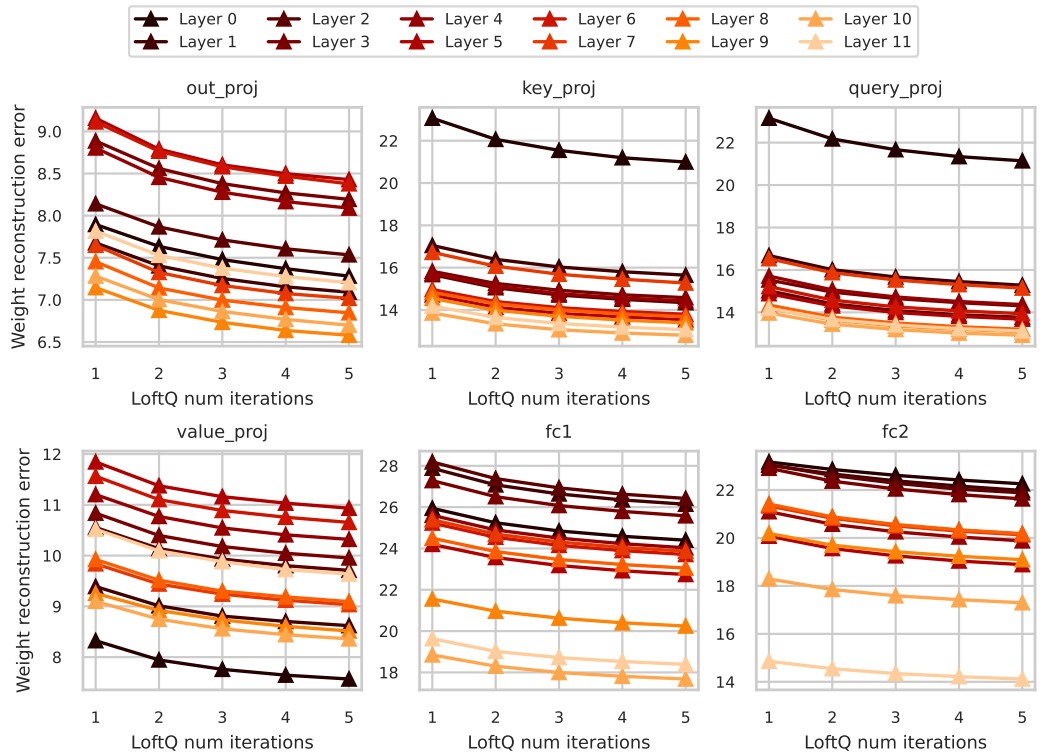

Figure 6: Weight approximation error of 3-bit rank-16 LoftQ with different numbers of iterations on RoBERTa-base. We observe that the weight reconstruction error of all the layers decreases as the number of iterations increases, but as shown in Figure 1b, the model output error ($k$=16) increases from the 4-th to 5-th iteration.

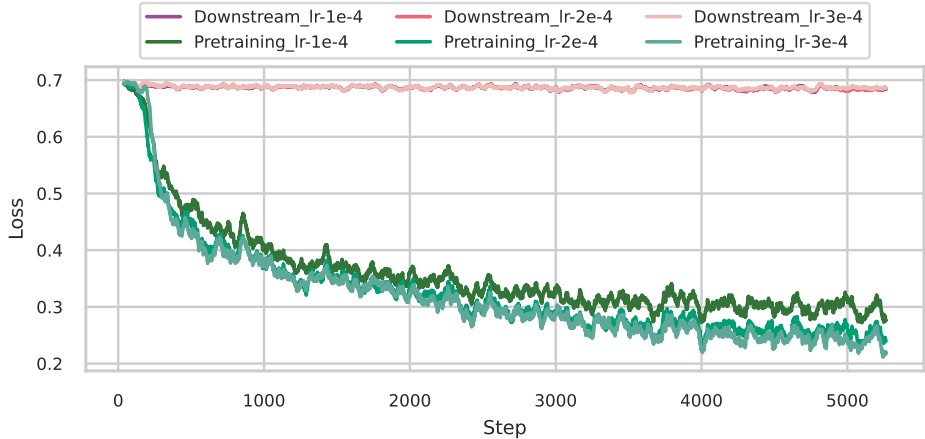

Figure 7: The fine-tuning loss curves of QERA-adapted 2-bit RoBERTa-base on SST2. The loss fails to decrease if the calibration is performed on the downstream task SST2 due to the massive padding tokens in preprocessed SST2 samples. In pretraining dataset, there are only a few special tokens like padding tokens and mask tokens.

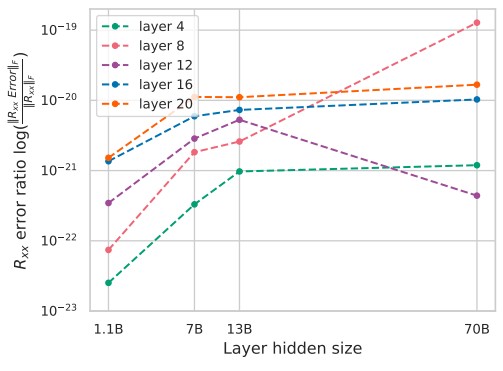

(a) Estimated error ratio of the square root of $\mathbb{R}_{\mathrm{XX}}$

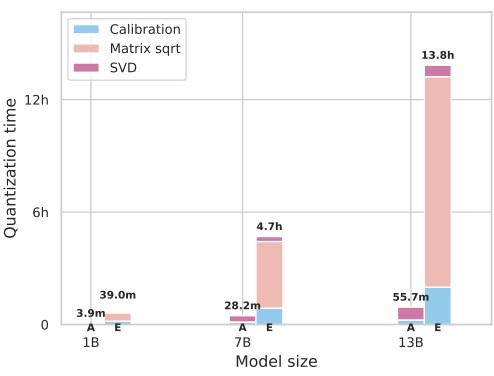

(b) QERA quantization time

Figure 8: Scalability of QERA. (a) plots the estimated error ratio of the matrix square root calculation of $\mathbb{R}_{\mathrm{XX}}$ of some layers where the error increases as the model goes larger. (b) compares the quantization time of QERA-approx and QERA-exact if all layers are quantized sequentially. The matrix square root is time-consuming since it is executed on CPUs. One key optimization for accelerating the quantization process of QERA-exact will be the GPU-accelerated matrix square root.

## A.7 SCALABILITY AND NUMERICAL STABILITY OF QERA

One may notice the diminishing model performance improvement of QERA-exact over QERA-approx as the model size increases. The main reason is that larger LLMs are more resistant to quantization (Chee et al., 2024). Another reason can be the error ratio of the matrix square root calculation of the autocorrelation matrix increases with model hidden size (Figure 8a).

We find that the data type used in the calibration is important for the numeric stability of QERA-exact due to the calculation of the matrix square root and SVD. To improve the stability of the calculation in QERA-exact, a good practice we find is to perform the outer product of $\boldsymbol{R}_{\mathrm{XX}}$ in FP32, accumulated outer product in FP64, and calculate the matrix square root in FP64 using the blocked Schur algorithm (Deadman et al., 2012). Figure 8b illustrates the quantization time of QERA-approx and QERA-exact on the platform described in Appendix A.4 where the linear layers are quantized sequentially. QERA-exact is slow due to the calculation of matrix square roots on CPUs. GPU-accelerated matrix square root will be the key optimization to reduce the quantization time. Note that in QERA, the quantization of individual layers is independent, allowing more parallelization and acceleration of the quantization process.

## A.8 CHOICE OF SOLUTIONS FOR QPEFT AND PTQ

QPEFT and PTQ are two different application scenarios of QERA. We recommend QERA-approx for QPEFT and QERA-exact for PTQ. PTQ aims to recover the model performance as much as possible without re-training. For PTQ, it is desirable to recover more model performance even if it takes longer to compute low-rank terms. Note that the low-rank terms are pre-computed once offline. At inference time, QERA-exact introduces no overhead to the hardware since LQER, QERA-approx, and QERA-exact all takes the same form of $\boldsymbol{y} = \boldsymbol{x}(\widetilde{\boldsymbol{W}} + \boldsymbol{A}_k \boldsymbol{B}_k)$.

However, for QPEFT experiments, it is unreasonable to pay a long time for initializing the low-rank terms for the limited improvement in output approximation error (i.e., QERA-exact/CALDERA), because 1) fine-tuning can recover the error, and 2) instead of spending much time on initialization, increasing training steps or increasing the rank number brings more gain in the fine-tuned accuracy. We run controlled experiments to support this claim. In Table 7 and Table 8, we run QPEFT experiments of RoBERTA-base on MRPC and LLaMA-2-7B on SlimPajama respectively. Compared to QERA-exact (Caldera's Lemma 4.2), QERA-approx achieves better accuracy/perplexity while taking $\frac{2}{3} \sim \frac{1}{2}$ of the time.

Table 7: Runtime comparison of QERA-exact and QERA-approx on MRPC. It is recommended using QERA-approx for QPEFT instead of QERA-exact.

| Method | Rank | Epochs | Init. time | Training time | Total time ($\downarrow$) | Acc. ($\uparrow$) |
|---|---|---|---|---|---|---|
| QERA-exact | 8 | 4 | 1.6min | 2.2min | 3.8min | 88.97 |
| QERA-approx | 12 | 4 | 21s | 2.2min | **2.6min** | **89.95** |
| QERA-approx | 8 | 5 | 21s | 2.7min | **3.1min** | **89.97** |

Table 8: Runtime comparison of QERA-exact and QERA-approx on SlimPajama. It is recommended using QERA-approx for QPEFT instead of QERA-exact.

| Method | Rank | Epochs | Init. time | Training time | Total time ($\downarrow$) | PPL. ($\downarrow$) |
|---|---|---|---|---|---|---|
| QERA-exact | 16 | 2 | 4.9h | 1.9h | 6.8h | 6.31 |
| QERA-approx | 64 | 2 | 29.6min | 2.1h | **2.6h** | **6.18** |
| QERA-approx | 16 | 4 | 28.2min | 4.0h | **4.5h** | **6.21** |

## A.9 CHOICES OF LORA RANKS, MODELS, AND PRECISIONS FOR QPEFT

**Rank = 8 for GLUE experiments** We notice LoftQ paper uses a large rank of 16 and 32 for fine-tuning on GLUE, which is larger than the commonly-used rank value of LoRA (4 or 8 in LoRA paper (Hu et al., 2021)). If we consider LoRA as the upper limit of QLoRA-like QPEFT methods (including LoftQ and QERA), to effectively compare these QPEFT methods, one easy way is to set the rank as the minimum value required by LoRA and check which QPEFT method achieves an accuracy closest to LoRA. This is why we choose rank = 8 for GLUE experiments (For 2-bit GLUE experiments we use a large rank 64 since the quantization is very aggressive). If we use rank = 32, LoRA and all the QPEFT methods may be over-parameterized and it will be hard to make a fair comparison in terms of fine-tuned accuracy. To support this claim, we sweep the rank of LoRA-adapted RoBERTA-base on SST2 and MRPC and show a large rank $k$ like 16 in LoftQ has over-parallelization problem in Table 9 and Table 10.

**RoBERTa *vs.* DeBERTa** When investigating the related work, we find that both RoBERTa and DeBERTaV3 (He et al., 2021) are used in QPEFT experiments (Guo et al., 2023; Li et al., 2023; Meng et al., 2024; Guo et al., 2023; Zhang et al., 2023). The reason why we chose RoBERTa is that the RoBERTa checkpoint on HuggingFace[2] is complete and compatible with both Hugging-Face's official examples of sequence classification[3] and masked language modeling[4]. Specifically, the RoBERTa checkpoint contains both the base model and the masked language modeling head but the DeBERTaV3's checkpoint[5] only contains the base model. As we know, the base model is enough for fine-tuning on downstream tasks. However, to calibrate on the pretraining dataset, we need the language modeling head to verify if our implementation of data preprocessing and calibration matches how the model was originally pretrained. Note that the quality of the statistic values in QERA like $\mathbb{R}_{XX}$ depends on the quality of the calibration set. Thus, without the language modeling head in the checkpoint, we cannot perform the QERA's calibration for DeBERTaV3 properly, ensure the correctness of statistics in QERA, and explore the effect of the choice of calibration sets.

## A.10 DETAILED PTQ RESULTS

Here we offer the detailed evaluation results for each downstream task in Tables 11 to 17.

## A.11 TEST OF ASSUMPTION 1

We provide more plots of normalized $\mathbb{R}_{XX}$ magnitude, $\frac{\mathrm{abs}(\mathbb{R}_{XX})}{\|\mathbb{R}_{XX}\|_F}$, across LLaMA-3.1-8B, LLaMA-2-7B, Mistral-7B-v0.3, and TinyLlama-1.1B in Figures 9 to 24, where dark pixels are elements close

---

[2]RoBERTa-base checkpoint: link

[3]HuggingFace example of sequence classification: link

[4]HuggingFace example of masked language modeling: link

[5]DeBERTaV3's checkpoint: link

Table 9: Over-parameterization problem. We sweep the rank $k$ of LoRA on SST2 and reported fine-tuned accuracy. The highest accuracy at rank $k = 12$ indicates over-parameterization happens for $k \geq 12$.

| Method | Rank $k$ | Learning rates | Best Acc. |
|---|---|---|---|
| LoRA | 4 | 1e-4/2e-4/3e-4/4e-4/5e-4/6e-4 | 94.38 |
| | 8 | 1e-4/2e-4/3e-4/4e-4/5e-4/6e-4 | 94.46 |
| | 12 | 1e-4/2e-4/3e-4/4e-4/5e-4/6e-4 | **94.73** |
| | 16 | 1e-4/2e-4/3e-4/4e-4/5e-4/6e-4 | 94.50 |
| | 20 | 1e-4/2e-4/3e-4/4e-4/5e-4/6e-4 | 94.50 |

Table 10: Over-parameterization problem. We sweep the rank $k$ of LoRA on MRPC and reported fine-tuned accuracy. The highest accuracy at rank $k = 12$ indicates over-parameterization happens for $k \geq 12$.

| Method | Rank $k$ | Learning rates | Best Acc. |
|---|---|---|---|
| LoRA | 4 | 1e-4/2e-4/3e-4/4e-4/5e-4/6e-4 | 87.99 |
| | 8 | 1e-4/2e-4/3e-4/4e-4/5e-4/6e-4 | 88.97 |
| | 12 | 1e-4/2e-4/3e-4/4e-4/5e-4/6e-4 | **89.95** |
| | 16 | 1e-4/2e-4/3e-4/4e-4/5e-4/6e-4 | 89.46 |
| | 20 | 1e-4/2e-4/3e-4/4e-4/5e-4/6e-4 | 89.71 |

to zeros. There are strongly correlated embedding channels in some k_proj and o_proj layers. The assumption fits better in MLP layers (gate_proj, up_proj, and down_proj), and holds for over 60% of the layers in LLMs.

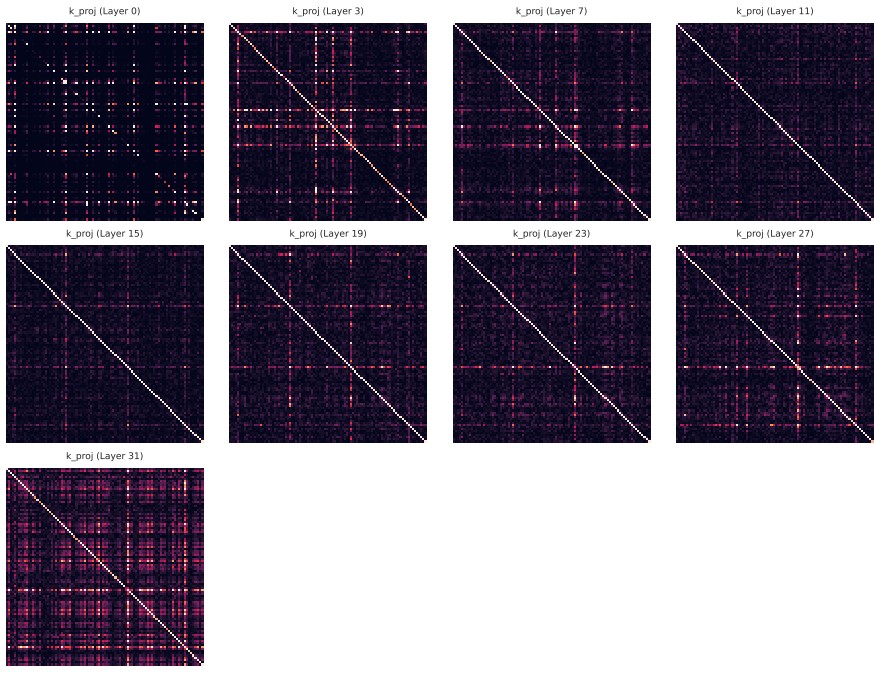

Figure 9: Normalized $\text{abs}(\mathbb{R}_{XX})$ of inputs of k_proj layers in LLaMA-3-8B. Note that the q_proj and v_proj share the same inputs. Layers are sampled and only the first 96 dimensions are plotted for clarity.

Table 11: Post-training quantization evaluation of TinyLlama-1.1B.

| rank | Method | w-bits | ARC (challenge) | BoolQ | CommonSenseQA | BBH | MMLU | WikiText2 | Winogrande |
|---|---|---|---|---|---|---|---|---|---|
| | | | Acc_norm | Acc | Acc | Acc_norm | Acc | Word ppl | Acc |
| - | BF16 | 16 | 32.51 | 55.93 | 20.07 | 29.68 | 25.35 | 13.98 | 59.59 |
| - | HQQ | | 32.00 | 58.13 | 20.15 | 29.70 | 25.75 | 15.02 | 59.35 |
| - | w-only | | 28.67 | 58.23 | 19.49 | 28.99 | 23.81 | 19.40 | 52.01 |
| | ZeroQuant-V2 | 4.25 | 29.69 | 57.86 | 19.41 | 29.53 | 24.85 | 18.03 | 52.57 |
| | LQER | | 32.00 | 52.42 | 18.59 | 29.60 | 25.31 | 16.23 | 59.75 |
| 32 | QERA-approx | | 31.83 | 52.08 | 17.20 | 29.51 | 25.22 | 15.66 | 58.72 |
| | QERA-exact | | 32.00 | 51.31 | 19.33 | 29.42 | 25.19 | 16.16 | 59.67 |

Table 12: Post-training quantization evaluation of Gemma-2-2B.

| rank | Method | W-bits | ARC (challenge) | BoolQ | CommonSenseQA | BBH | MMLU | WikiText2 | Winogrande |
|---|---|---|---|---|---|---|---|---|---|
| | | | Acc_norm | Acc | Acc | Acc_norm | Acc | Word ppl | Acc |
| - | BF16 | 16 | 49.91 | 72.60 | 50.29 | 32.67 | 49.44 | 13.08 | 68.82 |
| - | HQQ | | 48.81 | 71.77 | 48.40 | 32.32 | 46.52 | 14.29 | 67.40 |
| - | w-only | | 44.62 | 69.91 | 34.07 | 31.96 | 42.90 | 16.23 | 66.54 |
| | ZeroQuant-V2 | 4.25 | 44.45 | 69.94 | 34.07 | 31.50 | 43.27 | 15.71 | 66.22 |
| | LQER | | 46.08 | 68.84 | 37.59 | 32.60 | 45.78 | 14.55 | 67.72 |
| 32 | QERA-approx | | 45.31 | 68.99 | 36.20 | 32.04 | 45.80 | 14.60 | 67.40 |
| | QERA-exact | | 46.84 | 72.32 | 42.75 | 33.36 | 47.29 | 14.12 | 67.80 |

Table 13: Post-training quantization evaluation of Phi3-3.5-mini.

| rank | Method | W-bits | ARC (challenge) | BoolQ | CommonSenseQA | BBH | MMLU | WikiText2 | Winogrande |
|---|---|---|---|---|---|---|---|---|---|
| | | | Acc_norm | Acc | Acc | Acc_norm | Acc | Word ppl | Acc |
| - | BF16 | 16 | 59.39 | 84.65 | 71.91 | 48.19 | 64.58 | 11.50 | 72.77 |
| - | HQQ | | 57.00 | 74.34 | 60.20 | 38.22 | 56.00 | 14.63 | 69.61 |
| - | w-only | | 59.73 | 82.72 | 68.22 | 44.45 | 61.54 | 14.16 | 70.48 |
| | ZeroQuant-V2 | 4.25 | 59.64 | 82.94 | 68.06 | 44.58 | 62.00 | 14.09 | 69.77 |
| | LQER | | 59.39 | 84.01 | 70.76 | 45.67 | 62.21 | 12.88 | 70.74 |
| 32 | QERA-approx | | 59.45 | 84.82 | 70.84 | 45.67 | 62.26 | 12.81 | 70.17 |
| | QERA-exact | | 58.70 | 83.73 | 69.45 | 45.37 | 62.01 | 13.00 | 71.19 |

Table 14: Post-training quantization evaluation of LLaMA-2-7B.

| rank | Method | W-bits | ARC (challenge) | BoolQ | CommonSenseQA | BBH | MMLU | WikiText2 | Winogrande |
|---|---|---|---|---|---|---|---|---|---|
| | | | Acc_norm | Acc | Acc | Acc_norm | Acc | Word ppl | Acc |
| - | BF16 | 16 | 46.25 | 77.83 | 33.09 | 30.74 | 40.64 | 8.71 | 69.14 |
| - | HQQ | | 44.03 | 75.87 | 29.40 | 30.50 | 40.14 | 9.59 | 69.61 |
| - | w-only | | 45.22 | 75.87 | 25.47 | 30.71 | 40.03 | 9.45 | 68.43 |
| | ZeroQuant-V2 | 4.25 | 45.82 | 75.90 | 24.82 | 29.99 | 39.84 | 9.42 | 68.19 |
| | LQER | | 44.28 | 76.15 | 29.81 | 30.72 | 40.66 | 9.22 | 69.22 |
| 32 | QERA-approx | | 44.28 | 75.96 | 30.96 | 30.72 | 40.59 | 9.17 | 68.59 |
| | QERA-exact | | 44.80 | 76.39 | 31.61 | 30.57 | 40.86 | 9.12 | 69.22 |

Table 15: Post-training quantization evaluation of LLaMA-2-13B.

| rank | Method | W-bits | ARC (challenge) | BoolQ | CommonSenseQA | BBH | MMLU | WikiText2 | Winogrande |
|---|---|---|---|---|---|---|---|---|---|
| | | | Acc_norm | Acc | Acc | Acc_norm | Acc | Word ppl | Acc |
| - | BF16 | 16 | 49.49 | 80.58 | 47.34 | 32.65 | 52.18 | 7.68 | 72.22 |
| - | HQQ | | 49.06 | 78.69 | 45.05 | 32.41 | 50.85 | 8.27 | 71.11 |
| - | w-only | | 50.43 | 80.58 | 44.06 | 33.45 | 50.21 | 8.06 | 71.98 |
| | ZeroQuant-V2 | 4.25 | 50.00 | 81.04 | 44.47 | 33.50 | 50.31 | 8.07 | 71.59 |
| | LQER | | 51.02 | 81.25 | 44.47 | 32.41 | 51.24 | 7.96 | 71.98 |
| 32 | QERA-approx | | 51.11 | 80.83 | 44.06 | 32.48 | 51.07 | 7.95 | 71.67 |
| | QERA-exact | | 50.77 | 81.10 | 44.55 | 32.91 | 51.23 | 7.93 | 71.98 |

Table 16: Post-training quantization evaluation of LLaMA-3.1-8B.

| rank | Method | W-bits | ARC (challenge) | BoolQ | CommonSenseQA | BBH | MMLU | WikiText2 | Winogrande |
|---|---|---|---|---|---|---|---|---|---|
| | | | Acc_norm | Acc | Acc | Acc_norm | Acc | Word ppl | Acc |
| - | BF16 | 16 | 53.50 | 82.05 | 71.42 | 39.07 | 63.27 | 7.55 | 73.95 |
| - | HQQ | | 52.73 | 81.19 | 69.86 | 35.60 | 62.14 | 8.72 | 74.03 |
| - | w-only | | 50.68 | 81.31 | 67.24 | 37.34 | 59.03 | 8.78 | 73.56 |
| | ZeroQuant-V2 | 4.25 | 51.11 | 81.25 | 66.99 | 38.43 | 58.94 | 8.83 | 73.48 |
| | LQER | | 50.34 | 80.98 | 67.49 | 38.05 | 60.23 | 8.45 | 73.40 |
| 32 | QERA-approx | | 50.77 | 81.04 | 66.75 | 37.94 | 60.09 | 8.45 | 73.48 |
| | QERA-exact | | 51.28 | 80.18 | 68.83 | 37.48 | 60.60 | 8.33 | 73.95 |

Table 17: Post-training quantization evaluation of LLaMA-3.1-70B.

| rank | Method | W-bits | ARC (challenge) | BoolQ | CommonSenseQA | BBH | MMLU | WikiText2 | Winogrande |
|------|--------|--------|-----------------|-------|---------------|-----|------|-----------|------------|
|      |        |        | Acc_norm | Acc | Acc | Acc_norm | Acc | Word ppl | Acc |
| -    | BF16   | 16     | 65.10 | 85.38 | 78.46 | 48.53 | 75.28 | 3.06 | 79.56 |
| -    | HQQ    |        | 63.99 | 85.02 | 77.48 | 48.19 | 75.20 | 3.97 | 77.98 |
| -    | w-only |        | 60.58 | 83.82 | 73.63 | 41.28 | 73.06 | 4.55 | 78.37 |
| 32   | ZeroQuant-V2 | 4.25 | 59.90 | 83.61 | 73.55 | 42.75 | 73.15 | 4.48 | 77.74 |
|      | LQER   |        | 62.97 | 83.88 | 76.25 | 48.67 | 74.26 | 4.10 | 79.64 |
|      | QERA-approx |    | 62.12 | 83.79 | 76.74 | 48.53 | 73.98 | 4.10 | 79.64 |

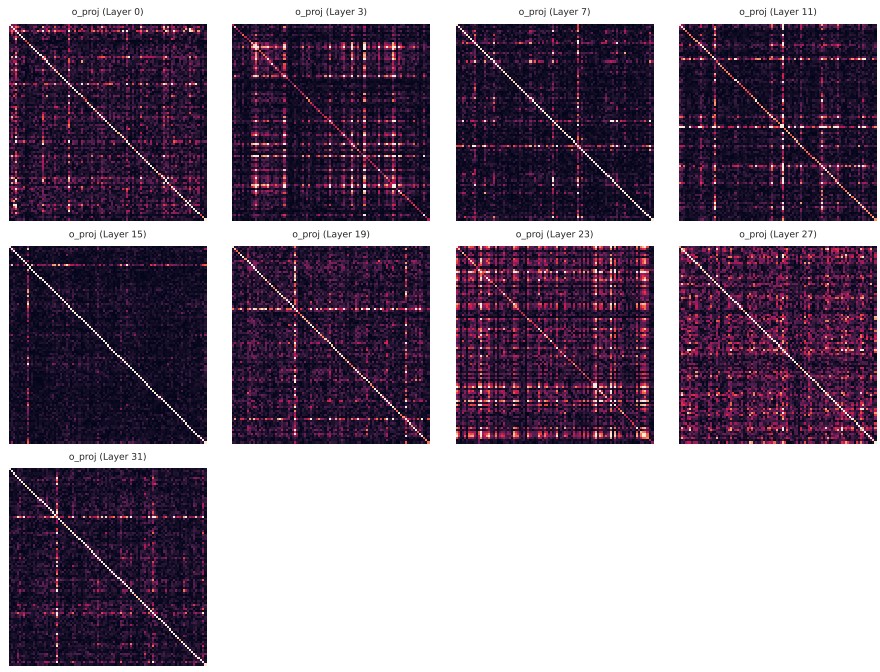

Figure 10: Normalized $\mathrm{abs}(\mathbb{R}_{\mathrm{XX}})$ of inputs of o_proj layers in LLaMA-3-8B. Layers are sampled and only the first 96 dimensions are plotted for clarity.

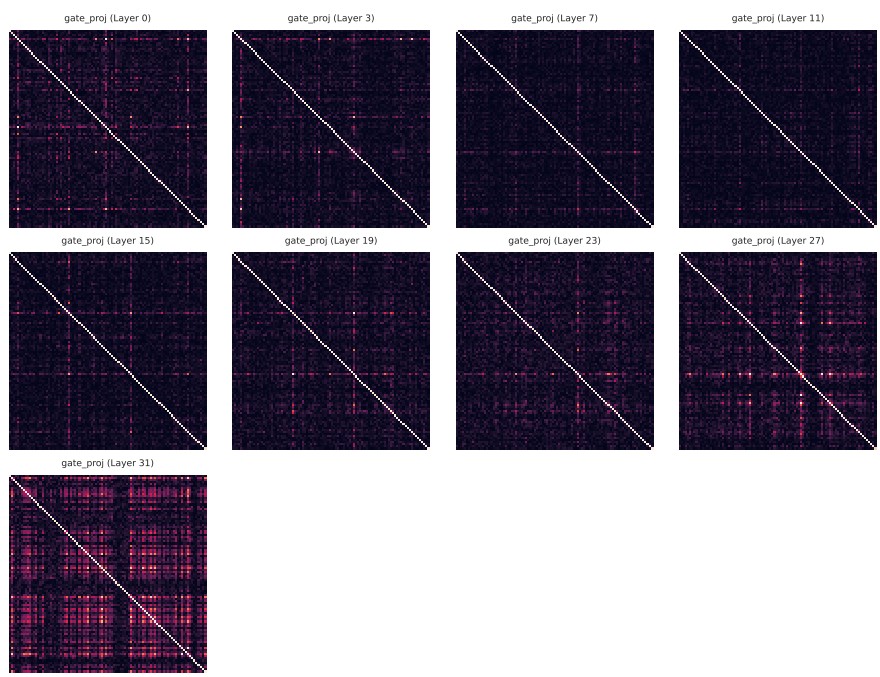

Figure 11: Normalized $\mathrm{abs}(\mathbb{R}_{\mathbf{XX}})$ of inputs of gate_proj layers in LLaMA-3-8B. Note that the up_proj shares the same inputs. Layers are sampled and only the first 96 dimensions are plotted for clarity.

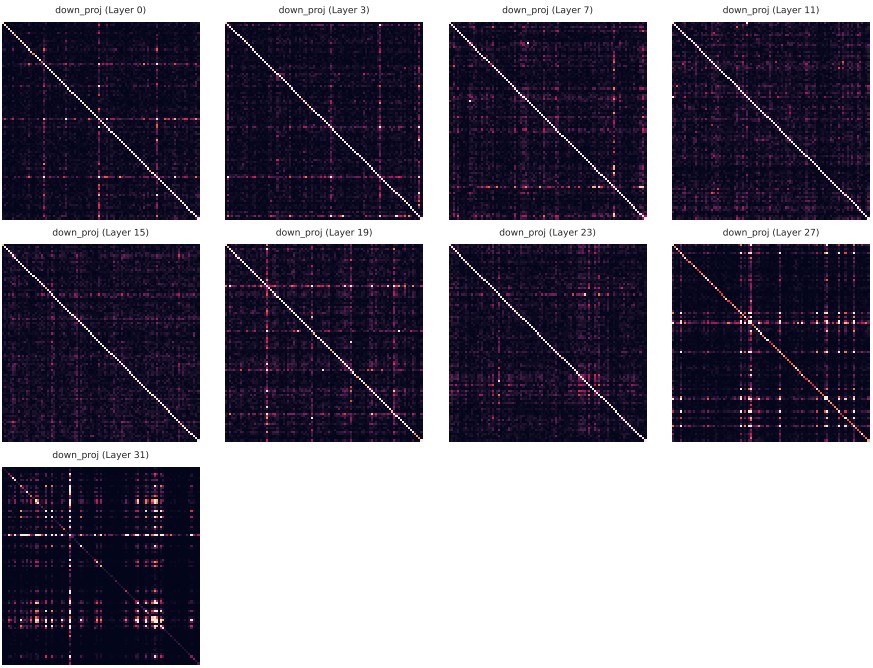

Figure 12: Normalized $\mathrm{abs}(\mathbb{R}_{\mathbf{XX}})$ of inputs of down_proj layers in LLaMA-3-8B. Layers are sampled and only the first 96 dimensions are plotted for clarity.

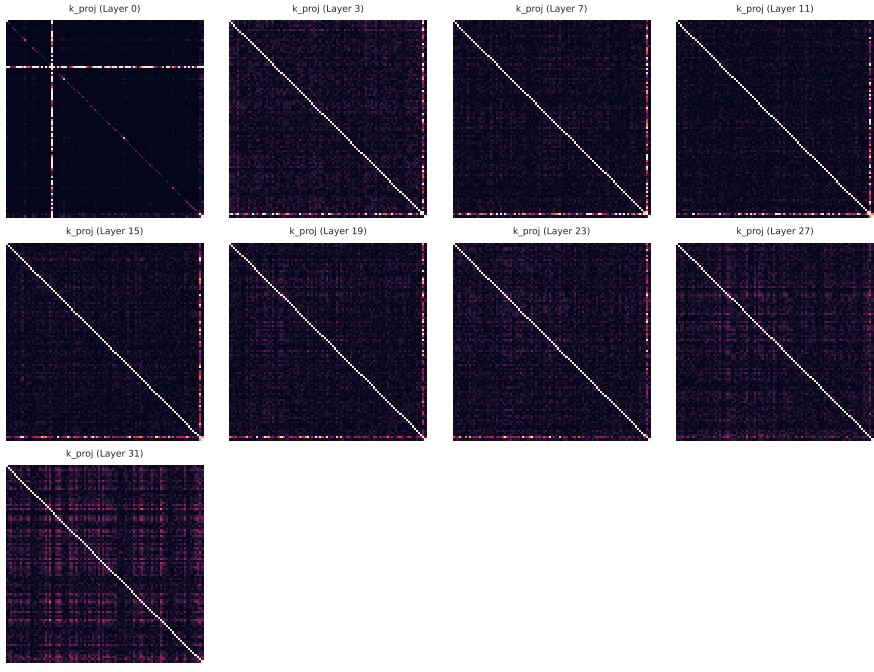

Figure 13: Normalized $\mathrm{abs}(\mathbb{R}_{\mathbf{XX}})$ of inputs of k_proj layers in LLaMA-2-7B. Note that the q_proj and v_proj share the same inputs. Layers are sampled and only the first 96 dimensions are plotted for clarity.

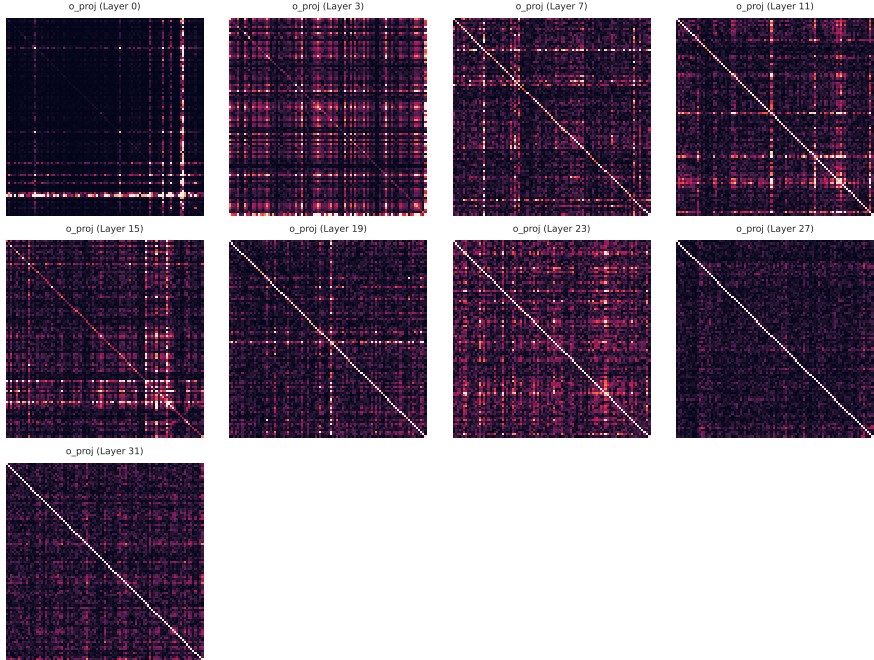

Figure 14: Normalized $\mathrm{abs}(\mathbb{R}_{\mathbf{XX}})$ of inputs of o_proj layers in LLaMA-2-7B. Layers are sampled and only the first 96 dimensions are plotted for clarity.

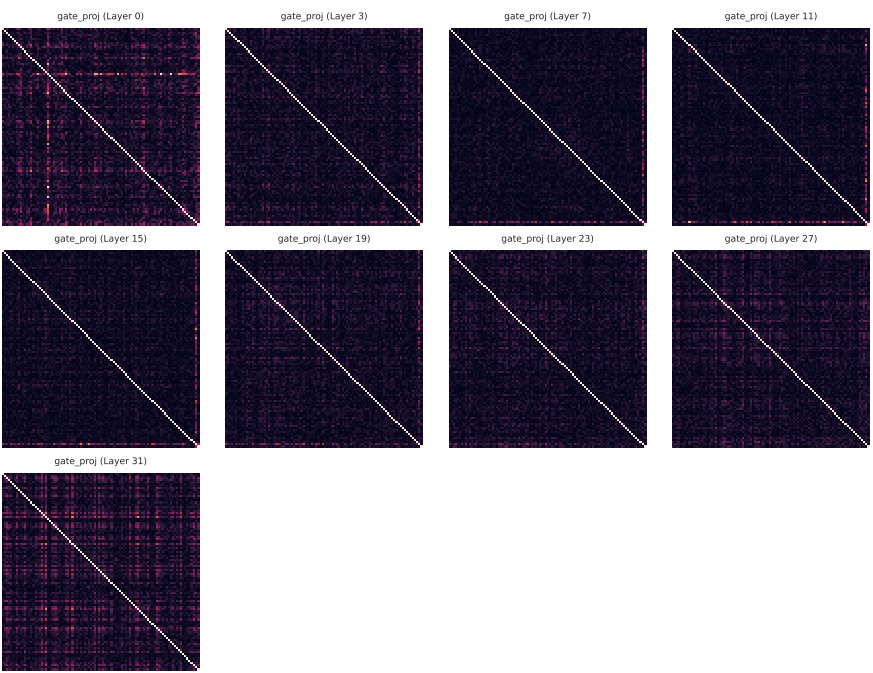

Figure 15: Normalized $\text{abs}(\mathbb{R}_{XX})$ of inputs of gate_proj layers in LLaMA-2-7B. Note that the up_proj shares the same inputs. Layers are sampled and only the first 96 dimensions are plotted for clarity.

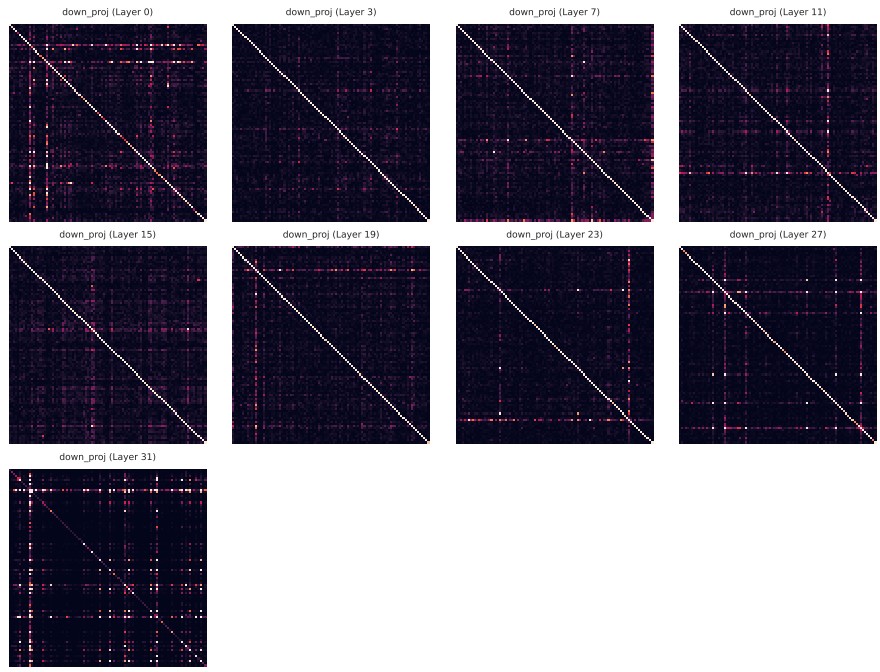

Figure 16: Normalized $\text{abs}(\mathbb{R}_{XX})$ of inputs of down_proj layers in LLaMA-2-7B. Layers are sampled and only the first 96 dimensions are plotted for clarity.

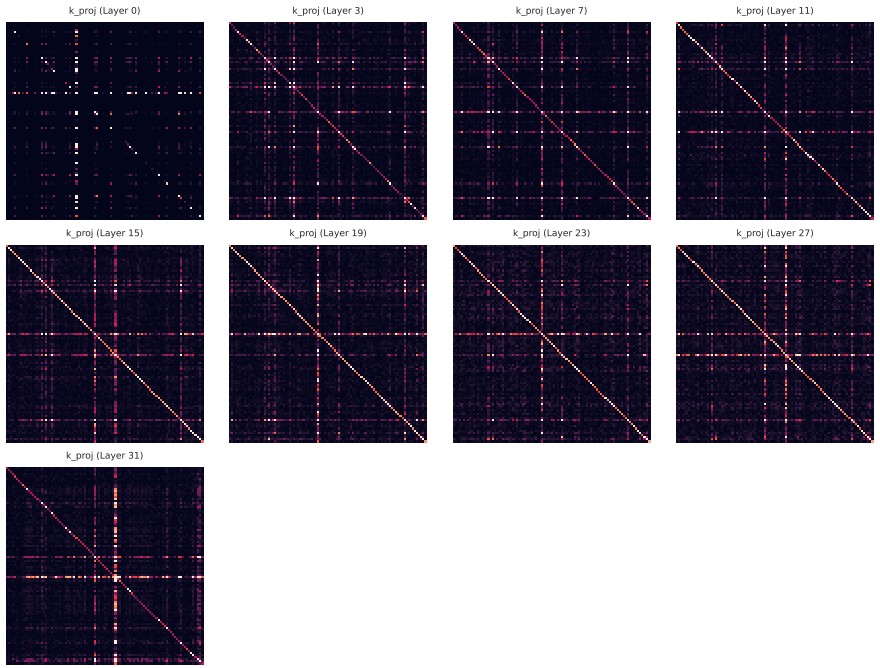

Figure 17: Normalized $\mathrm{abs}(\mathbb{R}_{\mathbf{XX}})$ of inputs of `k_proj` layers in Mistral-7B-v0.3. Note that the `q_proj` and `v_proj` share the same inputs. Layers are sampled and only the first 96 dimensions are plotted for clarity.

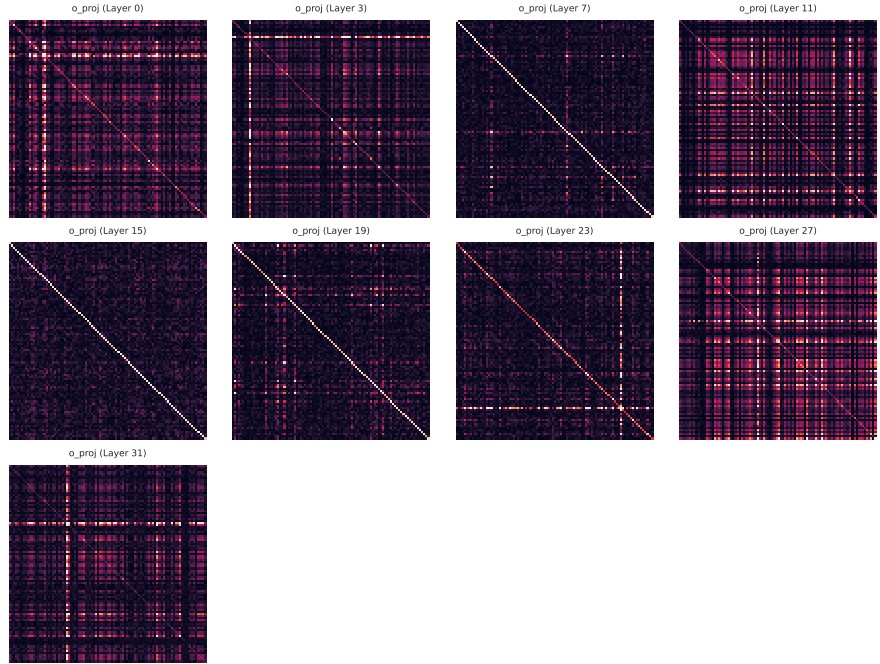

Figure 18: Normalized $\mathrm{abs}(\mathbb{R}_{\mathbf{XX}})$ of inputs of `o_proj` layers in Mistral-7B-v0.3. Layers are sampled and only the first 96 dimensions are plotted for clarity.

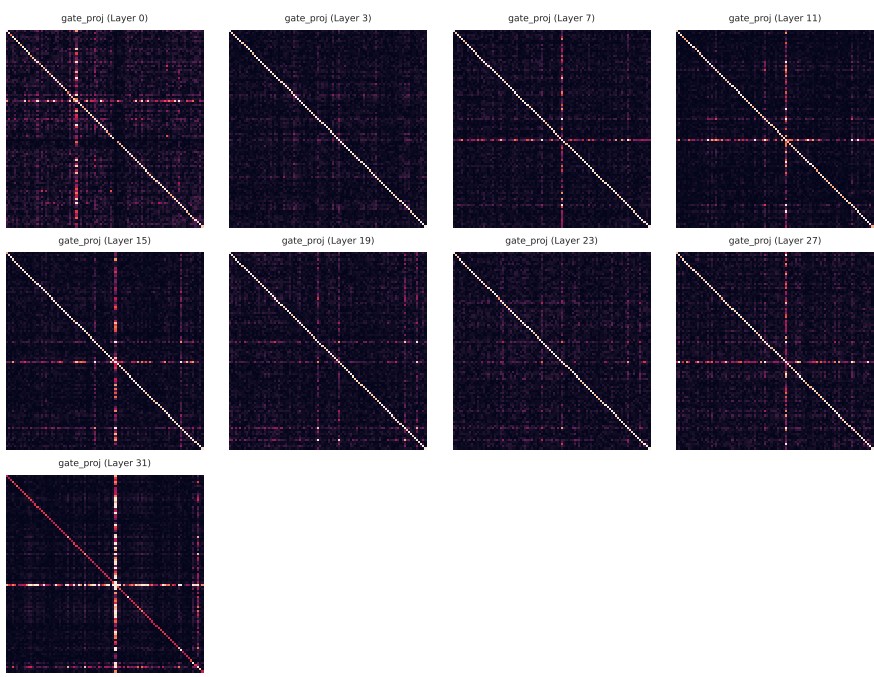

Figure 19: Normalized $\mathrm{abs}(\mathbb{R}_{\mathrm{XX}})$ of inputs of gate_proj layers in Mistral-7B-v0.3. Note that the up_proj shares the same inputs. Layers are sampled and only the first 96 dimensions are plotted for clarity.

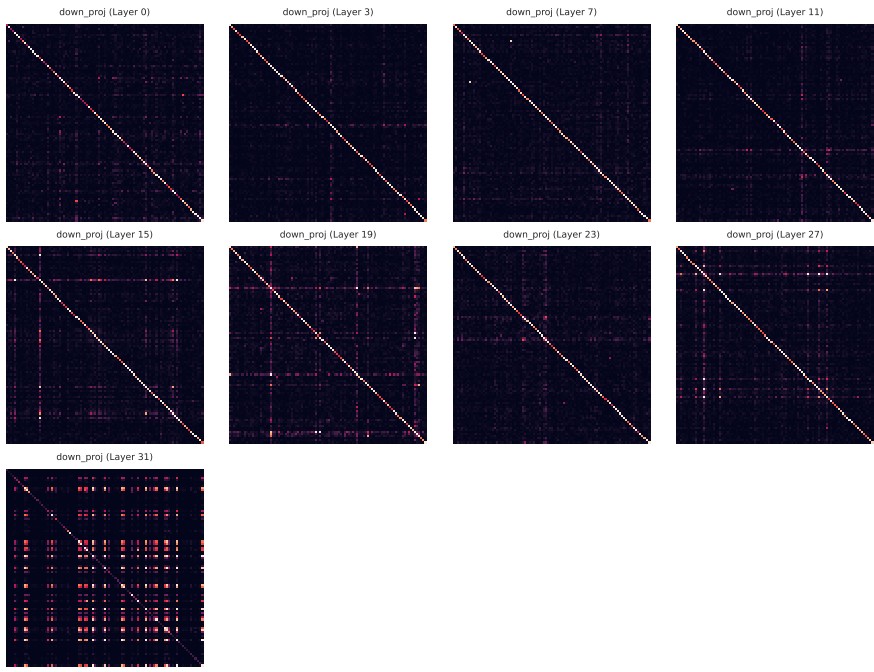

Figure 20: Normalized $\mathrm{abs}(\mathbb{R}_{\mathrm{XX}})$ of inputs of down_proj layers in Mistral-7B-v0.3. Layers are sampled and only the first 96 dimensions are plotted for clarity.

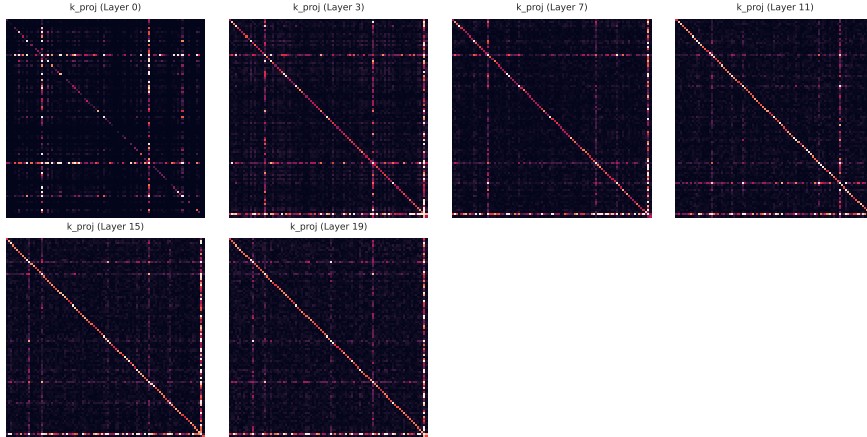

Figure 21: Normalized $\mathrm{abs}(\mathbb{R}_{\mathbb{XX}})$ of inputs of k_proj layers in TinyLlama-1.1B. Note that the q_proj and v_proj share the same inputs. Layers are sampled and only the first 96 dimensions are plotted for clarity.

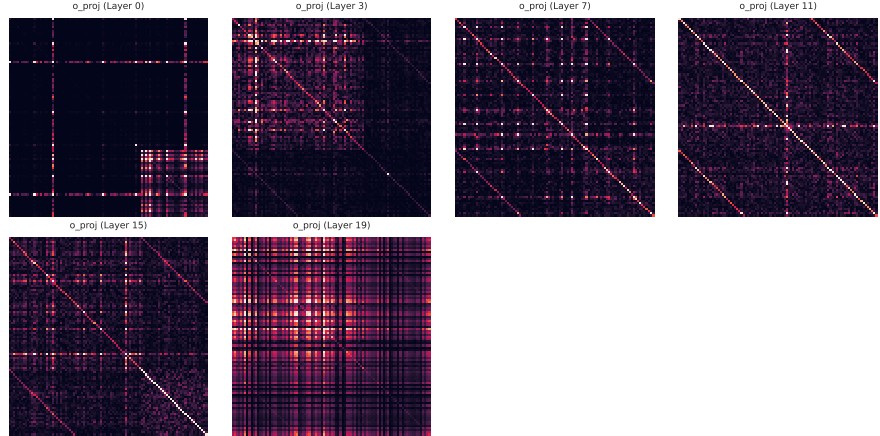

Figure 22: Normalized $\mathrm{abs}(\mathbb{R}_{\mathbb{XX}})$ of inputs of o_proj layers in TinyLlama-1.1B. Layers are sampled and only the first 96 dimensions are plotted for clarity.

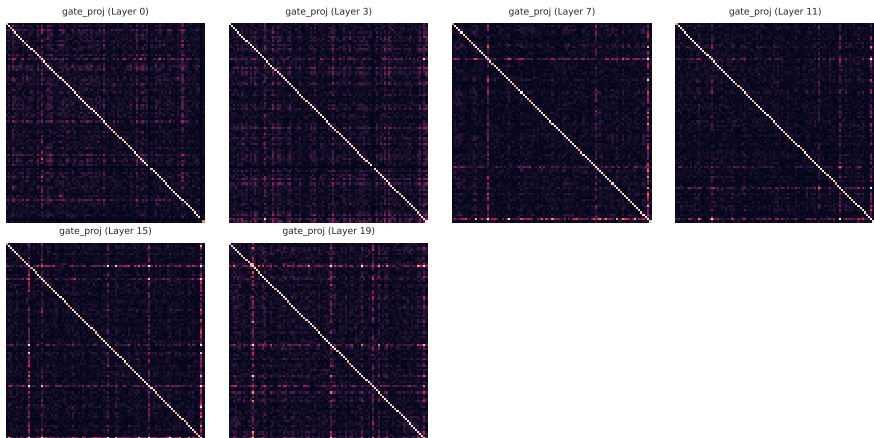

Figure 23: Normalized $\mathrm{abs}(\mathbb{R}_{\mathbb{XX}})$ of inputs of gate_proj layers in TinyLlama-1.1B. Note that the up_proj shares the same inputs. Layers are sampled and only the first 96 dimensions are plotted for clarity.

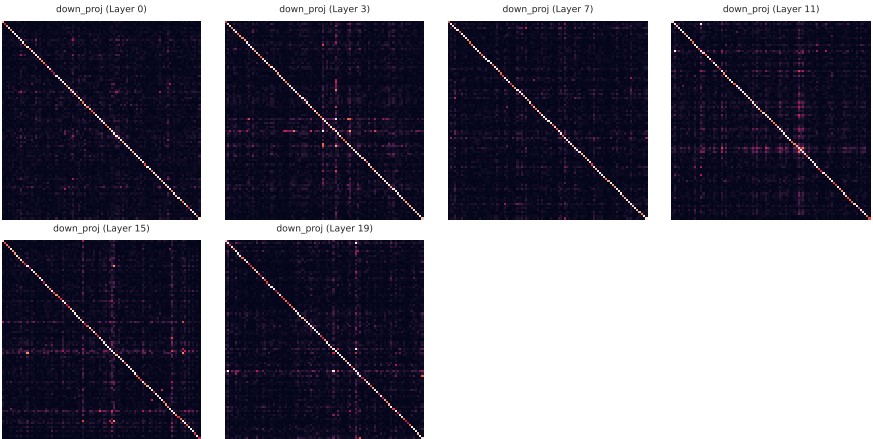

Figure 24: Normalized $\mathrm{abs}(\mathbb{R}_{\mathbb{XX}})$ of inputs of down_proj layers in TinyLlama-1.1B. Layers are sampled and only the first 96 dimensions are plotted for clarity.

