# OpenReview forum: "QERA: an Analytical Framework for Quantization Error Reconstruction"
_ICLR.cc/2025/Conference — ICLR 2025 Poster_

### Official Review · Reviewer_1yFJ · 2024-11-03

**Soundness:** 3
**Presentation:** 3
**Contribution:** 3
**Rating:** 6
**Confidence:** 4

**Summary:**

This paper proposes Quantization Error Reconstruction Analysis (QERA), an analytical framework for reconstructing quantization error with low-rank terms. QERA offers a closed-form solution to the problem of quantization error reconstruction and demonstrates that LLMs quantized with QERA achieve better linguistic capabilities compared to previous approaches.

**Strengths:**

1.	It introduces an analytical framework for reconstructing quantization error by considering the layer output, not just the weight values.
2.	It provides detailed mathematical proofs.

**Weaknesses:**

1.	Although the paper claims that demonstrating the relationship between minimizing layer output error and minimizing model output error is a key contribution, this claim seems overstated. Many existing quantization approaches [1, 2] for LLMs already consider layer output error rather than focusing solely on weight approximation error to achieve more accurate quantization.
2.	The overhead of considering layer output is not sufficiently discussed and is only briefly mentioned in Figure 12(b) of the appendix. This paper lacks a thorough analysis of the overhead (e.g., memory requirements and runtime) associated with the error reconstruction procedures. While weight approximation error has limitations in reconstructing errors, it offers more efficient procedures since it does not require a calibration dataset. However, approaches that consider layer input/output for error reconstruction do require a calibration dataset and more computation. Despite this discrepancy in calibration overhead, the paper does not provide a comprehensive analysis of the error reconstruction overhead in the proposed method compared to previous works.
3.	According to Figure 12(b) of the appendix, as QERA-exact appears to be very slow, it seems fair to compare LQER and QERA-approx. However, QERA-approx offers minimal advantage over LQER in terms of quantization.
4.	The paper does not compare the proposed method with LQ-LoRA, a more advanced method that also uses a calibration dataset for error reconstruction. A comparison between the proposed method and LQ-LoRA is important for evaluating QERA.
5.	The paper uses confusing terminology when categorizing experiments. It should use standard terms such as "quantization-aware fine-tuning" and "post-training quantization" to classify experiments, rather than "fine-tuning experiments" and "quantization experiments," as fine-tuning experiments also include quantization.

[1] Frantar, Elias, et al. "Gptq: Accurate post-training quantization for generative pre-trained transformers." ICLR 2023.

[2] Lin, Ji, et al. "AWQ: Activation-aware Weight Quantization for On-Device LLM Compression and Acceleration." MLSys 2024

**Questions:**

Please check the Weakness.

---

> ### Author Response · Authors · 2024-11-19
>
> # Q1
>
> Thank you for offering detailed suggestions and questions. Here we would like to answer them one by one. Meanwhile, the revised parts of the manuscript are colored in blue and the changes reflecting your comments are highlighted with **"@Reviewer 1yFJ" in pink**. The line numbers are based on the revised version.
>
> > Weaknesses:
> > 1. Although the paper claims that demonstrating the relationship between minimizing layer output error and minimizing model output error is a key contribution, this claim seems overstated. Many existing quantization approaches [1, 2] for LLMs already consider layer output error rather than focusing solely on weight approximation error to achieve more accurate quantization.
>
> Yes minimizing layer output error for quantization is not a new idea. However, the point we are shipping in the submission is that minimizing layer output error is more effective **for the quantization error reconstruction (QER) problem**, which takes the form of $\mathbf{y}=\mathbf{x}(W_q + A_kB_K)$, and we provide analytical solution to this problem.
>
> - We will revise the statement to avoid confusion or overstatement and carefully **limit our claim** in the context of quantization error reconstruction.
> - In line 140 page 3, we **already cited** activation-aware quantization works like **AWQ** to highlight the trend of minimizing output activation errors in PTQ area. Then we list recent quantization-error reconstruction works that uses activation-heuristics like **LQ-LoRA** in line 124 and line 154 page 3. Base on this summary of recent works, our statement is that the analytical solution to the quantization error reconstruction problem is **important but missing**.
> - The **reason** why we show this claim is because a series of works, [LRD (NeurIPS'21)](https://neurips2021-nlp.github.io/papers/27/CameraReady/Neurips_Workshop_camera_ready.pdf), , [ZeroQuant-v2(NeurIPS'22)](https://arxiv.org/pdf/2303.08302v3), [LPAF (COLING'24)](https://aclanthology.org/2024.lrec-main.945/), [LoftQ (the oral paper of ICLR'24)](https://openreview.net/forum?id=LzPWWPAdY4), may offer an inefficient way to the QER problem, that is, finding the $W_q$, $A_kB_k$ that minimizes the weight error. We use experiments in Fig.1 (line 324, page 7) and Fig.6 (line 918, page 16) to show the inefficiency of LoftQ.
> - Our **main contribution** is providing the exact and approximated analytical solutions to the quantization error reconstruction problem and showing our solutions are effective and useful, outperforming SoTA methods by a clear margin.

---

> ### Author Response · Authors · 2024-11-19
>
> # Q2-1
>
> > 2. The overhead of considering layer output is not sufficiently discussed and is only briefly mentioned in Figure 12(b) of the appendix. This paper lacks a thorough analysis of the overhead (e.g., memory requirements and runtime) associated with the error reconstruction procedures. While weight approximation error has limitations in reconstructing errors, it offers more efficient procedures since it does not require a calibration dataset. However, approaches that consider layer input/output for error reconstruction do require a calibration dataset and more computation. Despite this discrepancy in calibration overhead, the paper does not provide a comprehensive analysis of the error reconstruction overhead in the proposed method compared to previous works.
>
> Thank you for pointing out the missing analysis of the runtime and memory requirements of reconstruction procedures. We will **add the analysis and experiments** below to the revised version and show that QERA-approx is actually **more efficient** than LoftQ, the SoTA weight approximation method, for parameter-efficient fine-tuning.
>
> - First we clarify that the quantization time (the time of the reconstruction procedures) matters for parameter-efficient fine-tuning, but does not matter for post-training quantization.
>   - For parameter-efficient fine-tuning, the quantization time should be in a reasonable range because the fine-tuning experiments are usually fast compared to pre-training experiments, and may be rerun multiple times with tuned hyper-parameters like rank $k$ for better results. For example, when a new $k$ is picked, a new pair of $A_k$ and $B_k$ needs to be computed for initialization.
>   - For post-training quantization (inference), the quantization process is pre-computed once offline. At inference time, there is no difference to the hardware since all the quantization error reconstruction baselines take the same form of $\mathbf{y} = \mathbf{x}(W_q + A_k B_K)$. The aim of post-training quantization is to **recover the model performance as much as possible** without retraining. Therefore, it is desirable if the PTQ method recovers more model performance even if the quantization procedures take longer.
>   - As stated in line 346 page page 7, QERA-approx is for parameter-efficient fine-tuning (because it is effective enough and fast). QERA-exact is for post-training quantization as it has better model performance. We restress this point in the revised version in line 367, page 7 to avoid confusion.
>
> - Here we outline the computational complexity analysis we will add to the revised version.
>
>   Computational complexity of QERA
>
>   ```text
>   layer weight matrix size: (m, n)
>   rank: k
>   number of calibration samples: p
>   ============================================
>   QERA-exact complexity decomposition:
>   - Auto-correlation matrix Rxx: O(pm^2)
>   - Matrix square root of Rxx:   O(m^3)
>   - The inverse of Rxx^(1/2):    O(m^3)
>   - SVD on weights:              O(mn•min(m,n))
>   - The product to form Ck:      O(m^3)
>   ============================================
>   QERA-approx complexity decomposition
>   - Scale matrix S:                O(m^2•logm)
>   - SVD on weights:                O(mn•min(m,n))
>   - The product to form Ck:        O(m^3)
>   ```
>
>   Computational complexity of LoftQ
>
>   ```text
>   layer weight matrix size: (m, n)
>   number of iterations: t
>   LoftQ complexity decomposition
>   - Update quantized weight:       O(tmnk)
>   - Residual subtraction:          O(tmn)
>   - SVD on weights:                O(tmn•min(m,n))
>   ```
>
> - Here we show the runtime comparison between QERA-approx. and LoftQ.
>   - For LoftQ this means the runtime of the iterative SVD with 5-iterations (the official recommend setup).
>   - For QERA-approx this includes **both** the data calibration time with 128 samples and the computation required in Theorem 2 in line 302 page 6.
>
>   **Tab. R1: Comparison of runtime measured in seconds.** QERA-approx. is 2-4x faster than LoftQ.
>   |Method|1B| 3B|7B|13B|30B|
>   |---|---|---|---|---|---|
>   |LoftQ|170s|1037s |3068s|7080s| 45360s |
>   |QERA-approx| 70s | 263s | 738s | 1950s | 12495s |
>   |Speed up| 2.4$\times$ | 3.94$\times$| 4.16 $\times$ | 3.63$\times$ | 3.62$\times$ |
>
> - We attribute the larger overhead of LoftQ to the following reasons:
>   - The iterative SVD in LoftQ, an operation with the complexity `O(tmn•min(m,n))` per layer, is performed for all linear layers. In contrast, QERA-approx only performs the SVD once per layer.
>   - In each iteration, LoftQ updates both the weights and low-rank terms. This requires around `O(tmn)` memory read and write per layer between the GPU's global memory and local memory. For LLMs, this read and write is bottlenecked by the limited bandwidth of HBM. In contrast, QERA only calculates the low-rank terms, which means one `O(mn)` read for weights and one `O((m+n)k)` for low-rank terms write where the rank value `k` is very small (`k << min(m,n)`)
>
> Please refer to **Q2-2** for the remaining response to Q2

---

> ### Author Response · Authors · 2024-11-19
>
> # Q2-2
>
> - Here we analyze the memory footprint required by QERA and LoftQ
>
>   Memory footprint of QERA
>
>   ```text
>   layer weight matrix size: (m, n)
>   rank: k
>   batch size of calibration sample: b
>   ========================
>   QERA-exact:
>   - Calibration:
>     GEMM of linear forward pass: O(bm+mn+bn)
>     Partial Sum of Rxx:          O(m^2)
>   - Quantization:
>     Full-precision weights:      O(mn)
>     Quantized weights:           O(mn)
>     SVD on the scaled weight:    O(mn)
>     Matrix Square Root:          O(m^2)
>     Matrix Inverse:              O(m^2)
>     Low-rank terms:              O(mk+nk)
>   =======================
>   QERA-approx.:
>   - Calibration:
>     GEMM of linear forward pass: O(bm+mn+bn)
>     Partial Sum of S:            O(m)
>   - Quantization:
>     Full-precision weights:      O(mn)
>     Quantized weights:           O(mn)
>     SVD on the scaled weight:    O(mn)
>     Inverse of the scale matrix: O(m)
>     Low-rank terms:              O(mk+nk)
>   ```
>
>   Memory footprint of LoftQ
>
>   ```text
>   layer weight matrix size: (m, n)
>   number of iterations: t
>
>   LoftQ memory complexity:
>   Full-precision weights:   O(mn)
>   Quantized weights:        O(mn)
>   Low-rank terms:           O(mk+nk)
>   SVD on the scaled weight: O(mn)
>   ```
>
>   - QERA requires more memory than LoftQ. However, this should not be the problem because both LoftQ and QERA requires much less memory than the fine-tuning or the inference process of the adapted model, which are the purpose of applying LoftQ/QERA.
>
> All of the above analysis has been added to *A.8 Complexity Analysis* in the revised version.

---

> ### Author Response · Authors · 2024-11-19
>
> # Q3
>
> > 3. According to Figure 12(b) of the appendix, as QERA-exact appears to be very slow, it seems fair to compare LQER and QERA-approx. However, QERA-approx offers minimal advantage over LQER in terms of quantization.
>
> Here we would like to clarify a few factors that the reviewer may have ignored.
>
> - Though the quantization process of QERA-exact is slower than QERA-approx and LQER, this **does not influences deployment**. At inference time, there is no difference to the hardware since QERA-exact, QERA-approx, and LQER take the same form of $\mathbf{y} = \mathbf{x}(W_q + A_k B_K)$. The aim of post-training quantization is to **recover the model performance as much as possible** without retraining. Therefore, it is desirable that QERA-exact recovers more model performance even the quantization procedures take longer.
> - The quantization process of QERA-exact can be **accelerated**.
>   - As stated in the caption of the old Fig.12(b) (now it is moved to Fig.8(b)) and the appendix in line 1003 page 19, the runtime of QERA-exact showed in the barplot is much longer than QERA-approx is because we simply use the **SciPy's CPU implementation** of matrix square root, while both QERA-approx and LQER are implemented in pure GPUs. A GPU version of matrix square root will efficiently accelerate the process and is the key optimization to adapt QERA-exact for production. We did not explore the GPU implementation of matrix square root since we think this is **out of the scope of our paper**.
>   - As stated in line 1003, the barplot shows the runtime if all linear layers are quantized sequentially. Since there is no dependency between the quantization of different layers in QERA-exact, the runtime can be shorten by $N$ times if the layers are partitioned to $N$ chunks and quantized in parallel. This is actually how we run the experiments if the reviewer check the `ptq_pipeline_chunked.py` in the supplementary materials.
> - QERA-approx explains why the activation-induced heuristics in LQER works as a PTQ method, and it does not have the unstable problem in LQER since it is analytical. However, **for post-training quantization, we will recommend QERA-exact as it outperforms LQER by a clear margin** (See experiments in Section 4.3).
>     - As stated in line 143, page 3, LQER is activation-induced heuristics. We present experiments to show LQER is unstable in terms of the number of calibration samples in line 469, page 9. Our QERA-approx solves this problem and explains why the heuristics in LQER works as a PTQ method.
>     - Since the aim of PTQ is to recover the model performance as much as possible without retraining. QERA-exact is our recommended method for PTQ, which outperforms LQER by a clear margin in Tab.3 and Tab.4 on page 9. Again, we re-stressed that it is desirable to pay extra quantization time to recover more model performance, and the QERA-exact does not introduce any overhead to the deployment/inference compared to LQER or QERA-approx.
> - In summary, we think the quantization time of QERA-exact is not a fatal weakness of QERA-exact, and we do not encourage the reviewer to compare the model performance of QERA-approx with LQER in the PTQ setup as **QERA-approx is not for PTQ but for QPEFT**.

---

> ### Author Response · Authors · 2024-11-19
>
> # Q4
>
> > 4. The paper does not compare the proposed method with LQ-LoRA, a more advanced method that also uses a calibration dataset for error reconstruction. A comparison between the proposed method and LQ-LoRA is important for evaluating QERA.
>
> We would like to present the additional experiments to compare with LQ-LoRA. Besides the experiment, we also discuss other advantages of QERA over LQ-LoRA.
>
> - We show QERA outperforms LQ-LoRA in the following PEFT and PTQ experiments.
>   - We present the parameter-efficient fine-tuning results comparing QERA-approx to LQ-LoRA in **Tab.R2**. QERA-approx consistently achieves better fine-tuned perplexity/accuracy on SlimPajama and GSM8K.
>
>     **Tab. R2: Parameter-efficient fine-tuning results on SlimPajama and GSM8K**. QERA-approx uses the same setup as the submission. For LQ-LoRA we use the [official implementation](https://github.com/HanGuo97/lq-lora).
>     | W-bits | Method | LLaMA-2-7B | LLaMA-2-7B | LLaMA-3.1-8B | LLaMA-3.1-8B |
>     | ---    | ---    | ---        | ---        | ---          | ---          |
>     |        |        |SlimPajama ($\downarrow$)  | GSM8K ($\uparrow$) | SlimPajama ($\downarrow$) | GSM8K ($\uparrow$) |
>     | 16     | LoRA   | 6.17       | 39.40      | 8.07         | 55.72        |
>     | 4.25   | LQ-LoRA     | 6.36  | 31.12      | 8.71         | 54.67        |
>     | 4.25   | QERA-approx |**6.33**|**32.26**  |**8.68**      |**55.24**     |
>     | 2.25   | LQ-LoRA     | 11.49 | 18.63      | 22.14        | 17.84        |
>     | 2.25   | QERA-approx |**10.56**| **18.78**    | **20.07**      |**19.41**|
>
>   - We present the post-training quantization results comparing QERA-exact to LQ-LoRA in **Tab.R3**. QERA-exact outperforms LQ-LoRA in most of the cases.
>
>     **Tab.R3: PTQ Perplexity ($\downarrow$) on WikiText2**
>     | W-bits | Method | TinyLlama | Gemma-2 | Phi-3.5 | LLaMA-3.1 |
>     | ---    | ---    | ---       | ---     | ---     | :--:       |
>     |        |        | 1.1B      | 2B      | 3.8B    | 8B        |
>     | 16     | BF16   | 13.98     | 13.08   | 11.50   | 7.55      |
>     | 4.25   | LQ-LoRA| **15.98** | 14.76   | 13.43   | 8.64      |
>     | 4.25   | QERA-exact | 16.16 | **14.12**|**12.30**|**8.33**  |
>     | 3.25   | LQ-LoRA|     20.38 | 21.53   | **19.15**   | 11.77     |
>     | 3.25   | QERA-exact | **19.51** | **19.97**   | 20.37   | **11.39** |
>
> - **QERA is the mathematically optimal** solution to minimizing layer output error, In contrast, **LQ-LoRA is "ultimate heuristic"** (The author of LQ-LoRA uses "ultimate heuristic" to describe it. See the "Discussion and Limitations" section of [LQ-LoRA paper](https://arxiv.org/pdf/2311.12023)).
>   - As stated in the related work section (line 124, page 3), LQ-LoRA uses activation-induced heuristics to compute $W_q$, $A_k$, and $B_k$. Specifically, similar to LoftQ, LQ-LORA uses iterative SVD to update these terms, and early exits the iterations when the activation heuristic term $||D_{row} (W - W_q - A_kB_K) D_{col}||_F^2$ stops decreasing on a calibration set.
>   - Heuristic methods may introduces more hyper-params like the number of iterations in LQ-LoRA and have potential pitfalls when applied to real word applications like the problems of LoftQ and LQER we showed in the submission. In contrast, QERA is neat and guaranteed by math proof.
>
> - The quantization procedure of QERA-approx is faster than LQ-LoRA when adopted for parameter-efficient fine-tuning. We have shown that QERA-approx is faster than LoftQ. On top of LoftQ, LQ-LoRA further introduces extra cost of calculating Fisher information and tracking heuristic objective for early exit.
>
> The additional experiments and analysis have been added to *A.11 Comparison to LQ-LoRA* in the revised version.

---

> ### Author Response · Authors · 2024-11-19
>
> # Q5
>
> > 5. The paper uses confusing terminology when categorizing experiments. It should use standard terms such as "quantization-aware fine-tuning" and "post-training quantization" to classify experiments, rather than "fine-tuning experiments" and "quantization experiments," as fine-tuning experiments also include quantization.
>
>
> Sorry for the confusion. We will correct the usage of terms in the revised version. **We will try our best to make the revised version clear** enough such that the reader can easily follow it.
>
> - We assume the reviewer refers to the paragraphs of "fine-tuning experiments" and "post-training quantization experiments" in the Experiment Setup Section in line 346 and 355. We will rename the paragraphs in the revised version to "QPEFT experiments" and "PTQ experiments". The definition of QPEFT and PTQ is already stated in line 45 and line 49 page 1.
> - Throughout the Introduction Section and the Related Work Section in the submission, we clearly split the application scenarios of quantization error reconstruction problem into Parameter-Efficient Fine-Tuning and Post-Training Quantization separately (e.g., line 44 page 1, line 96 page 2, line 129 page 3). For convenience, in line 45 page 1, we noted them as QPEFT and PTQ respectively and highlight that QPEFT is for training and PTQ is for inference. In Section 4 Experiments, we also split the experiments into "4.2 Improved QPEFT" and "4.3 Improved PTQ", presented results and analyzed them separately. **We intended to distinguish them throughout the paper to help the reader follow the context**.
> - If the reviewer finds more confusing expressions/statements, we are happy to fix them and we welcome further discussion.
>
> The terms have been corrected in the revised manuscript.

---

> ### Author Response · Authors · 2024-11-21
> **Reaching the End of the Public Discussion Phase**
>
> Dear Reviewer,
>
> We have added experiments, and analysis to answer your questions and hope that our responses have sufficiently addressed the concerns you raised. We welcome further discussion if you have more questions and suggestions.
>
> As the discussion deadline is approaching, we would be very grateful if you could take a moment to review our reply. Thank you for your time and consideration.

---

> ### Author Response · Authors · 2024-11-23
>
> Dear Reviewer,
>
> As the rebuttal is closing in 3 days, we would be grateful if you could kindly read our reply to your questions and advices.
>
> We addressed your concerns with additional clarifications, experiments and analysis, and highlighted the revised part in the manuscript with "@Reviewer 1yFJ" in pink.
>
> If you have follow-up questions, we will try our best to answer them. Thank you for your time and consideration.

---

> > ### Comment · Reviewer_1yFJ · 2024-11-25
> >
> > Thank you for the detailed reply. After carefully reviewing the responses and revisions, I have decided to increase my score.

---

### Official Review · Reviewer_KXUq · 2024-11-03

**Soundness:** 4
**Presentation:** 3
**Contribution:** 3
**Rating:** 8
**Confidence:** 3

**Summary:**

This paper directly gives the analytic solutions of the linear compensation matrix for quantization error correction.  The analytic solutions involve both exact and approximate solutions to tradeoff compensation quality and efficiency. The analytic solutions are solidly validated for two scenarios, post-training and fine-tuning-based quantization.

**Strengths:**

1. The work formally demonstrates that optimizing to reduce the model layer output error is more effective than minimizing the weight quantization error. Given the fundamental importance of the optimization objective in this field, the work has a significant impact.

2. The assumption used to approximate the exact solution appears reasonable and can be validated through practical testing.

3. The experiments conducted in the work are extensive and comprehensive.

**Weaknesses:**

1. The efficiency of the exact and approximate solutions, like computational complexity and the concrete execution time compared with related works, should be further clarified.
2.  Eq. (18), $\mathop{\arg\min}_{Q_k} $.
3. Can the analytic solution be used in LLM pruning? Given that pruning and quantization usually have similar mathematical modeling to some extent.

**Questions:**

Please see the weaknesses.

---

> ### Author Response · Authors · 2024-11-19
>
> # Q1
>
> Thank you for the valuable suggestions and positive feedback! We would like to answer your questions one by one. Meanwhile, the revised parts of the manuscript are colored in blue and the changes reflecting your comments are highlighted with **"@Reviewer KXUq" in gray**. The line numbers are based on the revised version.
>
> > Q1. The efficiency of the exact and approximate solutions, like computational complexity and the concrete execution time compared with related works, should be further clarified.
>
> - Yes we notice this common concern pointed out by the reviewers. Here we present the computational complexity analysis.
>
>   QERA
>
>   ```text
>   layer weight matrix size: (m, n)
>   rank: k
>   number of calibration samples: p
>   ============================================
>   QERA-exact complexity decomposition:
>   - Auto-correlation matrix Rxx: O(pm^2)
>   - Matrix square root of Rxx:   O(m^3)
>   - The inverse of Rxx^(1/2):    O(m^3)
>   - SVD on weights:              O(mn•min(m,n))
>   - The product to form Ck:      O(m^3)
>   ============================================
>   QERA-approx complexity decomposition
>   - Scale matrix S:                O(m^2•logm)
>   - SVD on weights:                O(mn•min(m,n))
>   - The product to form Ck:        O(m^3)
>   ```
>
>   LoftQ
>
>   ```text
>   layer weight matrix size: (m, n)
>   number of iterations: t
>   LoftQ complexity decomposition
>   - Update quantized weight:       O(tmnk)
>   - Residual subtraction:          O(tmn)
>   - SVD on weights:                O(tmn•min(m,n))
>   ```
>
> - In our fine-tuning experiments, we noticed that the execution time of QERA-approx is around 2-4$\times$ faster than LoftQ.
>   - For QERA the execution time includes data calibration on 128 samples and calculating low-rank terms using SVD.
>   - For LoftQ the execution time refers to the iterative SVD to calculate quantized weights and low-rank terms.
>   - This is mainly because the iterative SVD in LoftQ is computation intensive, and LoftQ updates $W_q$ in every iteration, where requires extra `O(tmn)` memory write operations. We will also add the following **Tab.1** to the revised version for a straightforward runtime comparison.
>
>   **Tab. R1: Comparison of execution time**. The runtime of quantization procedures measured in seconds.
>   | Method | 1B | 3B | 7B | 13B | 30B |
>   | --- | --- | --- | --- | --- | --- |
>   | LoftQ  | 170s | 1037s | 3068s | 7080s | 45360s |
>   | QERA-approx | 70s | 263s | 738s | 1950s | 12495s |
>   | Speed up  | 2.4$\times$ | 3.94$\times$| 4.16 $\times$ | 3.63$\times$ | 3.62$\times$ |
>
> We have added the computational complexity as well as memory complexity and the runtime experiments to *A.8 Complexity Analysis*.
>
> - The Fig.12 in the submission appendix on page 19 offers an overview of the execution time of QERA-approx and QERA-exact. Since we use Scipy's CPU implementation of matrix square root, the matrix square root calculation becomes the bottleneck. A GPU-accelerated matrix square root will be the key optimization to speedup the execution time.
>     - Note that for post-training quantization inference, this does not matter because the low-rank terms are precomputed once offline.
>     - For parameter-efficient fine-tuning we show QERA-approx is efficient and effective enough.

---

> ### Author Response · Authors · 2024-11-19
>
> # Q2
>
> > 2. Eq. (18),  $argmin_{Q_k}$
>
> Thank you for careful and patient reading! We have fixed this typo in the revised version.

---

> ### Author Response · Authors · 2024-11-19
>
> # Q3
>
> > 3. Can the analytic solution be used in LLM pruning? Given that pruning and quantization usually have similar mathematical modeling to some extent.
>
> Yes, this is definitely possible. If the approximated weight $\widetilde{W}$ in the objective Eq(9) in line 200 page 4, $\text{argmin}_{C_k} E\{||x(\widetilde{W}+C_k)-xW||_2^2\}$, is obtained from pruning, QERA becomes the drop-in solution. This is because the proof staring from line 233 page 5 has no constraints on $\widetilde{W}$, i.e., $\widetilde{W}$ can be obtained from any approximation methods, quantization, pruning, etc.
>
> One more interesting fact we would like to note is that If we assign $\widetilde{W}$ in Eq(9) to 0, our proof gives the optimal solution to this problem: Find the optimal low-rank approximation of weights, $C_k$, that minimizes the layer output error. This variant extends QERA to classic problem of low-rank compression of weights (no quantization here).

---

> > ### Comment · Reviewer_KXUq · 2024-11-19
> > **Thanks for reply**
> >
> > Thanks for the authors' reply. After reading the reply, I choose to retain my score.

---

### Official Review · Reviewer_tFvD · 2024-11-04

**Soundness:** 3
**Presentation:** 3
**Contribution:** 3
**Rating:** 8
**Confidence:** 4

**Summary:**

In this paper, authors give the  analytical solution to solve the minimizing the errors in layer outputs via low rank terms. And authors prove that  minimizing the layer output error is better than minimizing the weight approximation error from the aspect of model performance.

**Strengths:**

This article is excellent in aspect of motivation, problem solving, and paper writing, and is also highly recommended for  its algorithm engineering work.
1. In terms of motivation, this article chooses to use theoretical methods to solve problems that can only be solved using heuristic algorithms at this stage, and determines the theoretical extreme value of the problem and the method to reach it.

2. This article provides a very solid analytical method and gives an algorithm for solving the extreme value according to this method, which is solid and reliable.

3. Paper writing aspect, this article has been detailed and concise, simplifying the repetitive proof of Theorem 2 and placing the proof of Theorem 1 as an important part of the text, allowing readers to fully understand the contribution of this article. In addition, the structure of this article is clear, and the questions and answers are clearly stated.

4.The biggest advantage of this article is that it reasonably engineers the algorithm and provides a reasonable simplified algorithm that is easy to implement in engineering.

**Weaknesses:**

The authors insight that minimizing the output error is better than weight approximation error is is consistent with our practical experience in the aspect of model performance. However, this point is hard to prove via experiments, because we cannot enumerate all weight approximation methods on every models. The conclusion is so strong.
So, two suggestions are that 1. give a mathematical proof of this point. 2. avoiding discuss this conclusion in paper, and only show your work is better than SOTA low rank methods.

**Questions:**

1.How to prove that the reason of worse performance of weight approximation methods is rooted in the model's nature instead of your weight approximation methods choice in experiments?

2.In first paragraph, k<<min(m,n). I think it should be k << $\frac{mn}{m+n}$ because the computation cost of W is mn, the computation cost of $A_KB_K$is mk+nk, we want (m+n)k<mn, k should satisfy k <$\frac{mn}{m+n}$

---

> ### Author Response · Authors · 2024-11-19
>
> # Response to Weakness
>
> Thank you for the valuable suggestions. We would like to answer your questions one by one. Meanwhile, the revised parts of the manuscript are colored in blue and the changes reflecting your comments are highlighted with **"@Reviewer tFvD" in purple**. The line numbers are based on the revised version.
>
> > Weaknesses:
> > The authors insight that minimizing the output error is better than weight approximation error is is consistent with our practical experience in the aspect of model performance. However, this point is hard to prove via experiments, because we cannot enumerate all weight approximation methods on every models. The conclusion is so strong. So, two suggestions are that 1. give a mathematical proof of this point. 2. avoiding discuss this conclusion in paper, and only show your work is better than SOTA low rank methods.
>
> - Yes we only tested this claim of reducing layer output error by minimizing layer output error via experiments and show it works in our setup. This is the limitation of empirical claims, and also why we run both parameter-efficient fine-tuning and post-training quantization experiments to show the results support our claim in various models, tasks, rank values, and number formats.
> - Unfortunately, we do not think it is straightforward to prove the claim through a math proof as neural networks are blackbox models. The bridge between layer output error and the model output is missing.
> - We will carefully revise the submission to avoid strong claims, limit the discussion in the setup we run, and emphasize its limitation in the last section.

---

> > ### Comment · Reviewer_tFvD · 2024-11-23
> >
> > I think you should delete the claim that quantizing the minimizing the layer output error is better than the  minimizing the weight approximation error in all paper, for example, your contributions part.
> > A suggestion, I  present that "we propose a minimizing the layer output error method, and our minimizing the layer output error method is better than current SOTA minimizing the weight method"

---

> > > ### Author Response · Authors · 2024-11-23
> > >
> > > Thank you for offering detailed and actionable suggestions. Following your suggestions, we just revised and uploaded the manuscript.
> > >
> > > We **removed or modified** the statements in line 59 page 2, line 77 page 2, line 423 page 8 to avoid the over general claim of quantizing the minimizing the layer output error is better than the minimizing the weight approximation error, and **limit** the comparison/conclusion related to weight approximation method to the relative improvement over LoftQ method.
> > >
> > > If you have more suggestions or questions, we are glad for further revision and discussion.

---

> ### Author Response · Authors · 2024-11-19
>
> # Q1
>
> > Q1. How to prove that the reason of worse performance of weight approximation methods is rooted in the model's nature instead of your weight approximation methods choice in experiments?
>
> Sorry we do not follow your question. Could you elaborate the meaning of weight approximation method in your question?
>
> If your question is **if QERA works only with specific quantization methods (number formats)**. The answer is no.
> - As defined at the start of our proof (line 233, page 5), QERA does not have any assumption or constraints on the quantized weight $\widetilde{W}$. This statement is also re-stressed at the end of the proof (line 287, page 6), i.e., the approximated $\widetilde{W}$ can be obtained from any quantization methods, or even other approximation methods like pruning.
> - Our solution (QERA-exact, Theorem 1 in line 216, page 4) is the optimal solution to minimizing layer output error (Problem 2 in line 193, page 4) with only the invertibility assumption (Remark 1, line 227, page 5).
> - In contrast, for the heuristic and iterative method in LoftQ, there is no math guarantee that it will reduces the layer weight approximation error of any number formats, which agrees with our experiment in Fig.1 (page 7).
>
> If your question is **if the worse performance of LoftQ in Fig.1 page 7 is because we choose LoftQ as the weight approximation method**. Here are our clarifications.
> -  When the iteration number of LoftQ is 1, the LoftQ becomes the **mathematically optimal** solution of *Problem 1 Minimization of weight error* in line 174 page 4. This is the solution guaranteed by math that minimizes the weight error using low-rank terms (See *Solution to Problem 1* in line 184, page 4), and it is also the method used by ZeroQuant-v2. In Fig.1, we show that this gives worse model performance than QERA, which is the mathematically solution minimizing the layer output error.
> - When the iteration number of LoftQ is larger than 1, the LoftQ becomes an **empirical method** that minimizes the weight output error by updating both quantized weights and low-rank terms. Compared to iteration number = 1, this allows updating the quantized weights. We sweep the iteration number and show that though the weight approximation error decreases with iterations (Fig. 6 page 18), the model output error of LoftQ may increase, and is always larger than QERA by a clear margin.
> - Thus, both the math optimal way of minimizing the weight approximation error and the LoftQ's empirical one have worse model performance in terms of model output error than QERA. This agrees with our experiments in which QERA-approx outperforms LoftQ in PEFT setup and QERA-exact outperforms ZeroQuant-v2 in PTQ setup both by a clear margin.
>
> If we misunderstand your question, we welcome further discussion and would like to offer more results and analysis

---

> ### Author Response · Authors · 2024-11-19
>
> # Q2
>
> > Q2. In first paragraph, k<<min(m,n). I think it should be $k\ll\frac{mn}{m+n}$ because the computation cost of W is mn, the computation cost of $A_k B_k$ is mk+nk, we want (m+n)k<mn, k should satisfy $k\ll\frac{mn}{m+n}$
>
> Thank you for careful reading and the comments on the bound of rank $k$. The inequation $k\ll\frac{mn}{m+n}$ has a tighter bound than $\text{min}(m, n)$, since $\frac{mn}{m+n}<\text{min}(m,n)$ for any positive integers. We have revised the submission to reflect this tighter bound derived from computation.
>
> We made lots of efforts to QERA and showed extensive experiments on proving its effectiveness, we hope it can contribute to the PEFT and PTQ research.

---

> ### Author Response · Authors · 2024-11-21
> **Reaching the End of the Public Discussion Phase**
>
> Dear Reviewer,
>
> We hope that our responses have sufficiently addressed the concerns you raised. We welcome more discussion if you have more questions and suggestions.
>
> As the discussion deadline is approaching, we would be very grateful if you could take a moment to review our reply. Thank you for your time and consideration.

---

> ### Comment · Reviewer_tFvD · 2024-11-23
>
> Thanks for the authors' reply. After reading the reply, I improve my score.

---

### Official Review · Reviewer_8H2Q · 2024-11-04

**Soundness:** 2
**Presentation:** 3
**Contribution:** 2
**Rating:** 6
**Confidence:** 4

**Summary:**

This paper proposes QERA that analytically finds the proper low rank terms by minimizing the layer output error. They show that minimizing the layer output error is more closely related to minimizing the model output error than minimizing the weight approximation error. They empirically demonstrate their method is better than other previous methods in both QPEFT and PTQ perspectives.

**Strengths:**

1. The paper is generally well-written and easy to follow.
2. The idea of deriving the analytical solution to the low rank terms by minimizing the layer output error is new.

**Weaknesses:**

The weaknesses of this paper mostly come from the experiment part.

1. The numbers in Table 1 and Table 2 don't match with the loftq original paper. Is that because you change the experimental setup? Could you please show your method outperforms loftq in their setup?
2. In the original loftq paper, they includes some experimental results about 2bit fine-tuning. Could you also show some results about 2bit fine-tuning?

**Questions:**

N/A

---

> ### Author Response · Authors · 2024-11-19
>
> # Q1-1
>
> Thank you for careful reviewing and raising questions about experiment setup and results. We would like to answer them one by one. Meanwhile, the revised parts of the manuscript are colored in blue and the changes reflecting your comments are highlighted with **"@Reviewer 8H2Q" in orange**. The line numbers are based on the revised version.
>
> > Q1. The numbers in Table 1 and Table 2 don't match with the loftq original paper. Is that because you change the experimental setup? Could you please show your method outperforms loftq in their setup?
>
> Yes we use a different experiment setup from LoftQ paper. We would like to **explain** why we use a different setup, then **also show experiments matching LoftQ** paper setup. We also add the following explanation and additional experiments to the manuscript and highlight them in purple.
>
> In Table 1, we reported results on RoBERTA instead of DeBERTA, and we adjusted hyper-params. This is because (1) the calibration process in QERA requires a **fully open-sourced model**. (2) our setup ensures a **better controlled experiments** to compare with baselines. These design choices have been explained in *A.8 Choices of LoRA Ranks and Models* in our submission. Here we elaborate with more details.
>
> (1) **Why RobERTA:** We uses RoBERTA for GLUE experiments and LoftQ paper uses DeBERTA for experiments. We did not choose DeBERTA because its checkpoint on HuggingFace is incomplete and we cannot verify if our implementation of data calibration for DeBERTA is correct.
>
> - As we know, the encoder-only BERT-like model is trained in two phases, pretraining and fine-tuning. The pretraining phase trains the backbone model and the language modeling head (`lm_head`). In the fine-tuning phase, the backbone model is chained with a randomly-initialized classifier head (`cls_head`) and these two parts are trained together on downstream tasks like GLUE.
> - For fine-tuning usually we just need the backbone model. However, for GLUE experiments in this submission, **QERA needs to calibrate on the pretraining data to collect statistics.** We need `lm_head` to check the correctness of our implementation, i.e., whether our data preprocessing & calibration matches how the backbone model was pre-trained.
>   - [The RoBERTA checkpoint](https://huggingface.co/FacebookAI/roberta-base) provides the weights of `lm_head`, and the RoBERTA model works seamlessly with [HuggingFace's implementation of language model pretraining](https://github.com/huggingface/transformers/tree/main/examples/pytorch/language-modeling#robertabertdistilbert-and-masked-language-modeling). This facilitates our controlled experiments that verifies our theorems.
>   - [The DeBERTA checkpoint](https://huggingface.co/microsoft/deberta-v3-base) only includes the backbone model without `lm_head`, and it does not work with HuggingFace's pretraining codes. This is also why the HuggingFace's official Inference API of DeBERTA gives wrong answer to the simple example question "Paris is the [MASK] of France." (See the Inference API example on the right of the [official checkpoint page](https://huggingface.co/microsoft/deberta-v3-base)). Therefore, we cannot perform rigorous fine-tuning experiments with DeBERTA to verify our theorem.
>
> (2) **Why hyper-params are adjusted:** When designing the fine-tuning experiments, we firstly include full fine-tuning (Full FT), LoRA, and QLoRA as baselines in our experiments because we may consider them as a series of methods bounded by the fine-tuned accuracy:
>
> $$\text{QLoRA < LoRA < Full FT}$$
>
> For example, Full FT is the upper bound of LoRA, i.e., with proper hyper-params, the best model trained by LoRA will not be better than the one trained by Full FT.
>
> Then our objective is to compare QERA with QLoRA and LoftQ, i.e., determining their order on the left of the inequation above:
>
> $$\text{x < y < z < LoRA < Full FT}$$
>
> where "$\text{x<y<z}$" is the order to be further determined by the results of fine-tuning experiments.
>
> Note that LoRA is still the upper bound of the quantized fine-tuning methods (QLoRA, LoftQ, QERA). However, the preliminary of this controlled experiments is "proper hyper-params". Besides the training hyper-params like learning rates, we need to ensure **none of these QPEFT methods are over-parameterized**, otherwise the converged accuracy after fine-tuning cannot clearly reveal the quality of fine-tuning methods, instead the accuracy heavily depends on other training hyper-parameters like batch size, learning rates, etc. To meet this preliminary requirement, we carefully choose the experiment setup, and results in different ones from LoftQ paper:
>
> Please refer to **Q1-2** for the remaining response to Q1

---

> > ### Author Response · Authors · 2024-11-19
> >
> > # Q2
> >
> > > Q2. In the original loftq paper, they includes some experimental results about 2bit fine-tuning. Could you also show some results about 2bit fine-tuning?
> >
> > Yes, we would like to show additional results in Tab.R4, but here are clarifications we need to present first.
> >
> > - **There is no strictly 2-bit setup in LoftQ paper**. The uniform float and normal float in LoftQ are fine-grained quantization in which a block of integers shares a single FP16 number. For example, in LoftQ paper and its [official implementation](https://github.com/huggingface/peft/blob/221965b7e140bdd47fbd04c131d56029e089902c/src/peft/utils/loftq_utils.py#L34), if the `block_size` is 32, and the integers are 2-bit, the equivalent bit-width is $2+16/32=2.5\text{ bits}$. The following table lists the equivalent bits of LoftQ's 2-bit format.
> >
> >   | Group size | N-bits |
> >   | ---        | ---    |
> >   | 32         | 2.5    |
> >   | 64         | 2.25   |
> >   | 128        | 2.125  |
> >
> > - In our RoBERTA's 2.5-bit experiments (Table 1 in the paper), the matthew correlation coefficient of the fine-tuned QLoRA and LoftQ are 0 and 3.43 on CoLA respectively, which indicates very poor performance. In contrast, QERA has 26.43 (still poor but better than LoftQ). Therefore we stopped further decreasing the bit-width as the 2.5-bit experiment is enough to show our advantage over LoftQ.
> > - Here we additionally show the 2.125-bit results in Tab.R5 to answer the reviewer's question. **In the 2-bit setup required by the reviewer, QERA still outperforms LoftQ** by a clear margin. Note that we do not recommend using this setup in practice since the model performance is poor.
> >
> >   **Tab.R5: 2-bit GLUE Experiments**. $^*$ means the CoLA is excluded when calculating the average since both LoftQ and QERA have 0 matthew correlation coefficient on this task.
> >   | Method | W-bits | MNLI | QNLI | RTE | SST | CoLA | QQP | STSB | Avg.$^*$ |
> >   | --- | --- | --- | --- | --- | --- | --- | ---| ---| ---|
> >   | LoftQ | 2.125 | 74.81 | 82.57 | 55.96 | 85.09 | 0  | 85.62  | 52.46 | 71.26  |
> >   | QERA  | 2.125 | **80.26** | **85.72** | 55.96 | **89.11** | 0  | **87.34** | **66.68** | **77.51** |
> >
> > The experiments and explanation above have been appended to *A.10 Choices of LoRA Ranks, Models, and Precisions For QPEFT* in the revised version.

---

> > > ### Comment · Reviewer_8H2Q · 2024-11-20
> > >
> > > Thank you for the clarification on why the authors use RoBERTa instead of DeBERTa and 2 bit fine-tuning setup. Also, I appreciate authors' work on additional experiments. My concerns are well addressed, and I raise my score from 5 to 6.

---

> ### Author Response · Authors · 2024-11-19
>
> # Q1-2
>
> - Rank $k$ is set to 8 for 3/4-bit GLUE experiments to avoid over-parameterization, aligning with the rank value in the GLUE experiments of the [LoRA paper](https://arxiv.org/pdf/2106.09685).
>   - In contrast, LoftQ uses a larger rank $k=16$. In Tab.R1 and Tab.R2 below we show that $k=16$ may lead to over-parameterization problem in the training of RoBERTA on GLUE.
>
>     **Tab.R1: Fine-tuned accuracy of LoRA-adapted RoBERTA-base on SST2**. The highest accuracy at rank $k=12$ indicates over-parameterization happens for $k\ge 12$.
>     |Method| Rank | Learning rates | Best acc. |
>     |--- | :-: | :-: | :-: |
>     |LoRA|  4 |1e-4/2e-4/3e-4/4e-4/5e-4/6e-4 | 94.38 |
>     |LoRA|  8 |1e-4/2e-4/3e-4/4e-4/5e-4/6e-4 | 94.46 |
>     |LoRA| 12 |1e-4/2e-4/3e-4/4e-4/5e-4/6e-4|**94.73**|
>     |LoRA| 16 |1e-4/2e-4/3e-4/4e-4/5e-4/6e-4 | 94.50 |
>     |LoRA| 20 |1e-4/2e-4/3e-4/4e-4/5e-4/6e-4 | 94.50 |
>
>     **Tab.R2: Fine-tuned accuracy of LoRA-adapted RoBERTA-base on MRPC**.
>     |Method| Rank | Learning rates | Best acc. |
>     |--- | :-: | :-: | :-: |
>     |LoRA|  4 |1e-4/2e-4/3e-4/4e-4/5e-4/6e-4 | 87.99 |
>     |LoRA|  8 |1e-4/2e-4/3e-4/4e-4/5e-4/6e-4 | 88.97 |
>     |LoRA| 12 |1e-4/2e-4/3e-4/4e-4/5e-4/6e-4 | **89.95** |
>     |LoRA| 16 |1e-4/2e-4/3e-4/4e-4/5e-4/6e-4 | 89.46 |
>     |LoRA| 20 |1e-4/2e-4/3e-4/4e-4/5e-4/6e-4 | 89.71 |
>
> - In our experiments the learning rate is tailored for each method (Full FT, LoRA, QLoRA, LoftQ, QERA) and GLUE subset. Before running the final experiments, we sweep the learning rates for each combination of `[method, subset]`, and select the best learning rate for that combination. Then we repeat training over random seeds using the best learning rate and report the averaged accuracy.
>   - In contrast, given a GLUE subset, LoftQ uses the same learning rate for all the baselines and LoftQ. In Tab.R3 below, we show that the learning rate optimal for LoftQ is not optimal for other baselines.
>
>     **Tab.R3: Fine-tuned accuracy of LoftQ/QLoRA-adapted RoBERTA on MRPC**. Each methods may prefer a different learning rate, thus we sweep learning rates and select the best one for each baseline.
>     |Method | Rank | Learning rates | Best LR | Best acc. |
>     | --- | :-: | :-: | :-: | :-: |
>     | LoRA  | 8 | 1e-4/2e-4/3e-4/4e-4/5e-4/6e-4 | 5e-4 | 89.99 |
>     | QLoRA | 8 | 1e-4/2e-4/3e-4/4e-4/5e-4/6e-4 | 3e-4 | 88.24 |
>     | LoftQ | 8 | 1e-4/2e-4/3e-4/4e-4/5e-4/6e-4 | 3e-4 | 88.24 |
>     | QERA  | 8 | 1e-4/2e-4/3e-4/4e-4/5e-4/6e-4 | 2e-4 | 88.97 |
>
> In Table 2 (page 8), we evaluated on LLaMA-3.1-8B as this is the **newest LLaMA** model and it was not released when LoftQ paper was accepted last year. We also use [SlimPajama-627B](https://huggingface.co/datasets/cerebras/SlimPajama-627B) as the language modeling set and and fine-tune on its subset. This is because the dataset used in LoftQ paper, WikiText2, is very small (<1M tokens), and usually included in the pretraining dataset of LLMs. Models fine-tuned on it is easy to have overfitting problem. In contrast, we use SlimPajama to avoid this problem. To meet reviewer's requirement, we still **show additional results matching LoftQ paper setup** on older LLaMA models and WikiText2 in Tab.R4 below. QERA still outperforms LoftQ.
>
> **Tab.R4: Fine-tuning results on LLaMA-2 matching LoftQ setup**.
> |Method | W-bits | LLaMA-2-7B | LLaMA-2-7B | LLaMA-2-13B | LLama-2-13B |
> | --- | --- | ---| --- | --- | --- |
> |     |     |  WikiText2 | GSM8K | WikiText2 | GSM8K|
> | LoRA| 16 | 5.02 | 39.37  | 5.09 | 46.4 |
> | LoftQ | 4.25 | 5.26 | 34.7 | 5.14 | 44.8 |
> | QERA  | 4.25 | **5.11** | **36.8** | **5.12** | **45.1** |

---

### Official Review · Reviewer_8w8V · 2024-11-06

**Soundness:** 3
**Presentation:** 3
**Contribution:** 2
**Rating:** 6
**Confidence:** 4

**Summary:**

This work proposes a framework for analyzing the quantization error after it is compensated using low-rank, high-precision terms. In such a decomposition, a weight matrix $\mathbf{W}$ is approximately split as $\mathbf{\tilde{W}} + \mathbf{A}_k\mathbf{B}_k$, where $\mathbf{\tilde{W}}$ is the quantized weight, and $\mathbf{A}_k\mathbf{B}_k$ is the low-rank compensation. This work aims to minimize the Frobenius norm error of the outputs of each layer (in contrast to just the weight quantization error), and propose closed-form solutions for the low-rank terms. The improved benefits are validated with numerical experiments of both encoder-only (RoBERTa) and decoder-only (LLaMa family) models. The experiments involve both post-training quantization (PTQ), as well as parameter-efficient fine-tuning (PEFT).

**Strengths:**

The paper analytically considers the problem of compensating the quantization error using low-rank high-precision components. The paper is generally well-written, although the work will benefit if it takes into account and compares with more recent works which takes into account the same problem (see weaknesses below).

The numerical experiments are comprehensive, and the results on a wide variety of models are presented. They are also compared with some other prior works, and show improved benefits.

**Weaknesses:**

My major concern with this paper is that it fails to take into account more recent works in this area, and justify how it compares with those works. The contribution of not really clear in light of a more recently proposed algorithm, Caldera (https://arxiv.org/abs/2405.18886) solves the optimization problem (9) optimally, i.e., the output error is minimized and closed form solutions for the low-rank factors are obtained (ref. Lemma 4.2 in the paper). Could the authors highlight the difference in their result of QERA-exact solution (Thm. 1) with Lemma 4.2 (Caldera)?

Furthermore, the autocorrelation matrix, $\mathbf{R}_{\mathbb{XX}}$ (which is also referred to as Hessians, because it is the Hessian of the quadratic loss in (9)), does need to be computed using a calibration dataset -- but this is a one-time cost. Additionally, the approximation in Assumption 1 for QERA-Approx approximates the autocorrelation matrix as a diagonal matrix -- this is not necessarily true as shown in Figs. 5, 7 and 8.

Secondly, minimizing error in layer outputs for PTQ is not really a recent idea as mentioned in the paper. It has been around for a few years now. See for example, https://arxiv.org/abs/2004.10568.

Despite the weaknesses mentioned here, the numerical evaluations on the models and the regimes of 3/4-bit quantization are most likely new.

**Questions:**

Please see the Weaknesses section. I would be happy to readjust my score if they are satisfactorily addressed.

**Details Of Ethics Concerns:**

None needed.

---

> ### Author Response · Authors · 2024-11-19
>
> # Q1-1
>
> Thank you for the careful review and enlightening questions! Here we answer them one by one. Meanwhile, the revised parts of the manuscript are colored in blue and the changes reflecting your comments are highlighted with **"@Reviewer 8w8V" in green**. The line numbers are based on the revised version.
>
> > - My major concern with this paper is that it fails to take into account more recent works in this area, and justify how it compares with those works. The contribution of not really clear in light of a more recently proposed algorithm, Caldera (https://arxiv.org/abs/2405.18886) solves the optimization problem (9) optimally, i.e., the output error is minimized and closed form solutions for the low-rank factors are obtained (ref. Lemma 4.2 in the paper). Could the authors highlight the difference in their result of QERA-exact solution (Thm. 1) with Lemma 4.2 (Caldera)?
>
> - We did not notice this concurrent work when we were working on QERA. We find this work was just accepted at NeurIPS'24 two moth ago, which is almost the same time when we submitted QERA.
> - After carefully reading through the Caldera paper, although written in a different form, we figure out that Caldera's Lemma 4.2 is equivalent to Theorem 1 in the submission. Here we briefly show the derivation of the equivalence:
>   - Before the derivation, for convenience, we remove the quantized weight term $\widetilde{W}$ from Problem 2 (line 193, page 4), which does not change the proof. Now the problem becomes *finding the optimal low-rank approximation of the weight matrix, $W_k:=C_k$ that minimizes the layer output error*.
>   - A list of Notation:
>     - Number of calibration samples $b$
>     - Layer input feature size $m$, output feature size $n$
>     - Calibration set $X$: $b\times m$
>     - A sample in the calibration set: $\mathbf{x}: 1\times m$
>     - Original layer weights: $W: m\times n$
>     - Layer output $Y: b\times n$
>     - Approximated rank-$k$ weight: $W_k: m\times n$
>     - SVD decomposition of $X$: $X=\text{SVD}(X) = U\Sigma V^T$
>     - Truncated rank-$k$ SVD: $\text{SVD}_{k}(\cdot)$
>   - Proof of the equivalence:
>     - First the objective of QERA (Problem 2 in 183 page 4) is equivalent to the objective of Caldera's Eq(5):
>
>       QERA: $\min_{W_k} E_\mathbf{x}\{||\mathbf{x}(W_k - W)||_2^2\}$
>
>       Caldera: $\min_{W_k}||X(W_k-W)||_F^2$
>
>       We notice that QERA's objective is equivalent to Caldera's objective scaled by a factor of constant $n$
>     - Then we show Theorem 1 is equivalent to Caldera's Lemma 4.2. **If OpenReview fails to render the latex equations below, please refer to *A.3 Connection and Difference between Caldera and QERA* on page 15**,
>
>       QERA-exact:  $W_k:= {(R_{XX}^{\frac{1}{2}})}^{-1} \mathrm{SVD}_k (R_X^{\frac{1}{2}} W)$, noted as REq(1)
>
>       Caldera's Lemma 4.2: $W_k':=V\Sigma \cdot\text{SVD}_k(U^TY)$, noted as REq(2).
>
>       First we show $(R_{XX}^\frac{1}{2})^{-1}$ in REq(1) equals to $V\Sigma$ in REq(2) scaled by a constant ${\sqrt{b}}$.
>
>       $R_{XX}=\frac{1}{b} (X^TX)=V\Sigma U^T U\Sigma V^T=V\Sigma^2 V^T$
>
>       $R_{XX}^\frac{1}{2}=\frac{1}{\sqrt{b}} \Sigma V^T$ `(*)`
>
>       $(R_{XX}^\frac{1}{2})^{-1}={\sqrt{b}}V\Sigma^{-1}$
>
>       Then we show the term inside REq(1)'s $\mathrm{SVD}_k(\cdot)$ equals to the term inside REq(2)'s $\mathrm{SVD}_k(\cdot)$  scaled by the constant $\frac{1}{\sqrt{b}}$
>
>       $U^TY = U^TXW = U^TU\Sigma V^T W = \Sigma V^TW={\sqrt{b}}R_{XX}^\frac{1}{2}W$, the last equality is from `(*)`
>
>       $R_{XX}^\frac{1}{2}W=\frac{1}{\sqrt{b}} U^TY$
>
>
>
>       Therefore, the two solutions are equivalent.
>
> Please refer to **Q1-2** for the remaining response to Q1.

---

> ### Author Response · Authors · 2024-11-19
>
> # Q1-2
>
> - We will add Caldera to the related work of the revised version, explain the connection between Caldera and QERA, and clarify the difference between our work and Caldera:
>   - Different problem setup:
>     - Caldera aims to solves the problem $\min_{W_q, A_{k,q}, B_{k,q}}||X(W - W_q - A_{k,q}B_{k,q}) ||_F^2$,
>
>       where all $W_q$, $A_{k,q}$, and $B_{k,q}$ are quantized and variables to optimize, and $X$ denotes the calibration samples
>     - QERA aims to solves the problem $\text{min}_{C_k}||\mathbf{x}(W-W_q-C_k)||_2^2$, where $W_q:=\text{quantize}(W)$ is predefined given a quantization method $\text{quantize}(\cdot)$, and the real-number (high-precision) low-rank $C_k:=A_kB_k$ is the only variable to optimize.
>   - We simplify QERA-exact and derive QERA-approx, which is a computationally-efficient approximated solution.
>     - QERA-approx is more suitable for parameter-efficient fine-tuing than QERA-exact/Caldera (We will presents experiments to support this claim in the answers to next question), outperforming all the baselines by a clear margin.
>     - QERA-approx also overcomes the pitfalls in existing methods (Fig.1 page 7, Fig.3 page 9), and explains why previous heuristic methods like LQER work.
>   - The optimization objective is similar (vector form vs. matrix form), final closed-form solution is equivalent, but the proof of QERA-exact is different from Caldera
>     - We start from minimizing the l2-norm of the error of output vector on the calibration set, transforming the expection of l2-norm error to F-norm.
>     - We leverage the property of auto-correlation matrix to derive the solution in a clear and simple way, though the solutions are equivalent.
> The above explanation has been added to *2 Related Work* and *A.3 Connection and Difference between CALDERA and QERA* in the revised version.

---

> ### Author Response · Authors · 2024-11-19
>
> # Q2
>
> > - Furthermore, the autocorrelation matrix, $\mathbf{R}_{XX}$
>  (which is also referred to as Hessians, because it is the Hessian of the quadratic loss in (9)), does need to be computed using a calibration dataset -- but this is a one-time cost. Additionally, the approximation in Assumption 1 for QERA-Approx approximates the autocorrelation matrix as a diagonal matrix -- this is not necessarily true as shown in Figs. 5, 7 and 8.
>
> We would like to answer this question from two perspectives.
> - **Test of Assumption 1** Though the Assumption 1 does not perfectly fit the actual statistics, QERA-approx still brings significant improvement over the baselines.
>   - As stated in line 535 page 10, there are dimensions strongly correlated with others, like `o_proj` layers, but as shown in Appendix A.5, the assumption still fits over 60% layers. Here we offer more $R_{XX}$ plots of various models to support this observation. These plots have been added to *A.13 Test of Assumption 1* in the revised version.
>     - Fig.R1: [Mistral-7b-v0.3](https://s2.loli.net/2024/11/19/mufL8lZhs5oD2tX.jpg)
>     - Fig.R2: [TinyLlama-1.1b](https://s2.loli.net/2024/11/19/jn5zW8APDXUiSfV.jpg)
>     - Fig.R3: [Llama-2-7b](https://s2.loli.net/2024/11/19/Gz5hwc3IlHLOEYo.jpg)
> - **The advantage of QERA-approx**. For parameter-efficient fine-tuning, the runtime/computation overhead matters.
>   - As we did in the submission, we split the application scenarios of QERA into two cases: parameter-efficient fine-tuning (PEFT) and post-training quantization (PTQ). For PTQ (inference/deployment), it is desirable to recover more model performance even if it takes longer to compute the low-rank terms, thus QERA-exact/Caldera is recommended in this case.
>   - However, for PEFT experiments, the long time to initialize the low-rank terms for the limited improvement in output approximation error (i.e., QERA-exact/Caldera) before the fine-tuning starts does not make much sense. **A computationally efficient initialization method is important for PEFT** because
>     - Fine-tuning experiments are usually fast compared to pretraining, and may be rerun multiple times to tune hyper-parameters like rank $k$. For example, when a new rank $k$ is picked, a new pair of $A_k$ and $B_k$ needs to be computed.
>     - Fine-tuning can recover the error easily;
>     - Instead of spending much time on initialization, increasing training epochs or increasing the rank number brings more gain in the fine-tuned accuracy, while saving much time.
>   - Here in Tab.R1 and Tab.R2 we show additional experiments that QERA-approx achieves better fine-tuning results than QERA-exact with only 2/3-1/2 of the time budget, simply using a larger rank number or more training epochs.
>
>     Tab.R1: Fine-tuned accuracy and runtime of RoBERTA-base on MRPC
>     | Method     | Rank     | Epochs |Init. time | Training time | Total time ($\downarrow$)| Acc. ($\uparrow$) |
>     | ---        | ---      | ---    | ---       | ---           | ---        | ---  |
>     | QERA-exact | 8        | 4      | 1.6min    | 2.2min        | 3.8min     | 88.97|
>     | QERA-approx| 12       | 4      | 21s       | 2.2min        | 2.6min     | 89.95|
>     | QERA-approx| 8        | 5      | 21s       | 2.7min        | 3.1min     | 89.71|
>
>
>     Tab.R2: Fine-tuned perplexity and runtime of LLaMA-2-7B on SlimPajama.
>     | Method     | Rank     | Epochs |Init. time | Training time | Total time ($\downarrow$)| PPL. ($\downarrow$) |
>     | ---        | ---      | ---    | ---       | ---           | ---        | ---  |
>     | QERA-exact | 16       | 2      | 4.9h      | 1.9h          | 6.8h       | 6.31 |
>     | QERA-approx| 64       | 2      | 29.6min   | 2.1h          | **2.6h**   |**6.18**|
>     | QERA-approx| 16       | 4      | 28.2min   | 4.0h          | **4.5h**   |**6.21**|
>
>     We already show that QERA-approx outperforms baselines by a clear margin in the submission. Therefore, QERA-approx is effective and suitable for PEFT.
>
> The experiments and the explanation above have been added to *A.9 Choice of Solutions for QPEFT and PTQ*, Tab.14 and Tab.15 in the revised version.

---

> ### Author Response · Authors · 2024-11-19
>
> # Q3
>
> > - Secondly, minimizing error in layer outputs for PTQ is not really a recent idea as mentioned in the paper. It has been around for a few years now. See for example, https://arxiv.org/abs/2004.10568.
>
> - Thank you for pointing out this. One of the observations we would like to highlight in QERA is that for **the problem of quantization error reconstruction**, which takes the forms of $\mathbf{y}=\mathbf{x}(\widetilde{W}+A_kB_k)$, minimizing output error is more effective than minimizing weight approximation error, and we offer analytic solutions, QERA, to that problem.
>   - The reason why we would like to address this is that [LoftQ](https://openreview.net/forum?id=LzPWWPAdY4), the oral paper in ICLR'24, focuses on minimizing weight approximation error. However we find LoftQ is not the effective way to solve this problem. Work in this domain is varied, with some research concentrating on weight error reconstruction (eg. [LRD (NeurIPS'21)](https://neurips2021-nlp.github.io/papers/27/CameraReady/Neurips_Workshop_camera_ready.pdf), [ZeroQuant-v2(NeurIPS'22)](https://arxiv.org/pdf/2303.08302v3), [LPAF (COLING'24)](https://aclanthology.org/2024.lrec-main.945/), [LoftQ (ICLR'24)](https://openreview.net/forum?id=LzPWWPAdY4)) and others on output error reconstruction. Currently, there is no consensus within the field of quantization regarding which type of error is the most advantageous to optimize. We show a head-to-head comparison between these via an empirical study to shed light on this issue. We use the model output error experiments in Fig.1 (line 324, page 7) and Fig.6 (line 918, page 18) to support our claim. We do not want to state that we are the first and only to consider output error reconstruction, but would like to point out and ship the useful insights that performance wise this would work better than weight error reconstruction objectives.
>   - We will carefully check and correct the statements in the revised version to avoid overstatement, and limited the claim in the context of quantization error reconstruction.
> - Here we would like to shortlist our contributions
>     - We formalize the quantization error reconstruction into two problems (Problem 1 and Problem 2 on page 4), and derives the solutions.
>       - The solution to Problem 1 is straightforward; we derives the exact solution to Problem 2, QERA-exact, though we just realized it agrees with the Lemma 4.2 in the concurrent work Caldera after the rebuttal starts.
>     - We also offer an approximated solution, QERA-approx, which is much more computationally efficient than QERA-exact.
>     - We run extensive experiments to show the advantage of QERA-approx and QERA-exact
>       - QERA-approx is suitable for fine-tuning, outperforming baselines by a clear margin; besides, we also show that
>         - QERA-approx overcomes the pitfall in LoftQ: Reduced layer weight error $\ne$ reduced model output error (line 406, page 8).
>         - QERA-approx overcomes the pitfall in LQER: Model performance vs. calibration set size (line 469, page 9).
>       - QERA-exact is suitable for post-training quantization, outperforming baselines by a clear margin. It has more improvement over the baselines when the quantization is more aggressive.
>
> We made lots of efforts to QERA and showed extensive experiments on proving its effectiveness, we hope it can contribute to the PEFT and PTQ research.

---

> ### Author Response · Authors · 2024-11-21
> **Reaching the End of the Public Discussion Phase**
>
> Dear Reviewer,
>
> We updated additional proof, experiments, and analysis to answer your questions and hope that our responses have sufficiently addressed the concerns you raised. We welcome more discussion if you have more questions and suggestions.
>
> As the discussion deadline is approaching, we would be very grateful if you could take a moment to review our reply. Thank you for your time and consideration.

---

> ### Author Response · Authors · 2024-11-23
> **Public Discussion Phase Ending Soon**
>
> Dear Reviewer,
>
> As the rebuttal is closing in 3 days, we would be grateful if you could kindly read our reply to your questions and advices.
>
> We addressed your concerns with additional proof, clarifications, experiments and analysis, and highlighted the revised part in the manuscript with "@Reviewer 8w8V" in green.
>
> If you have follow-up questions, we will try our best to answer them. Thank you for your time and consideration.

---

> > ### Comment · Reviewer_8w8V · 2024-11-26
> > **Response to rebuttal**
> >
> > Dear reviewer,
> >
> > Firstly, sincere apologies for responding so close to the end of the rebuttal period
> >
> > Thank you for your detailed response. I appreciate all the additional experiments done. I also agree with your justification for QERA-Approx.
> >
> > However, my major concerns remain, and I would appreciate it if you could respond to them briefly without diving deep into details.
> >
> > 1. Re Q-1: Comparison with Caldera: I agree with most of your observations regarding equivalence. But wouldn't you agree that Caldera is a general case of your work? Caldera considers the low-rank factors to be quantized, be one can relax this constraint. Then isn't the optimization problem same as Lemma 4.2 (which is special case of the optimization problem in Caldera)?
> >
> > See also https://dl.acm.org/doi/10.1145/2339530.2339609. Isn't eq. (3) essentially the same problem that you solve in QERA?
> >
> > Additionally, the proof of QERA-Exact is not very different from Caldera. You are considering the symmetric square root of the auto-correlation matrix (or Hessian). If I understand correctly, this is essentially a simplification of the proof of Lemma 4.2 in Caldera under the assumption that the auto-correlation matrix is PSD. Please correct me if I'm mistaken?
> >
> > 2. Re Q-3: Without the low-rank factors, isn't the optimization problem considered in your work the same as that in https://arxiv.org/abs/2004.10568? My point here pointing this out was merely the fact the minimizing the output error of a layer is not a very recent idea, and earlier references exist. See for example, the optimization problem of eq. (13) in https://arxiv.org/abs/2004.10568 exist. It is call it the *task-based loss*, but would you disagree that this is the same *output error minimization* formulation QERA considers?
> >
> > Please correct me again if I'm mistaken. I really appreciate the extensive experiments (including additional ones) that the authors did, but having a hard time distinguishing the novelty of the idea in the paper. Reiterating back,  I would highly appreciate it if you could respond to them briefly without diving deep into details.

---

> ### Author Response · Authors · 2024-11-26
>
> # Answers to Follow-up Questions (Part 1)
> Thank you for the follow up questions. Here are our answers.
>
> > Comparison with Caldera: I agree with most of your observations regarding equivalence. But wouldn't you agree that Caldera is a general case of your work? Caldera considers the low-rank factors to be quantized, be one can relax this constraint. Then isn't the optimization problem same as Lemma 4.2 (which is special case of the optimization problem in Caldera)?
> > See also https://dl.acm.org/doi/10.1145/2339530.2339609. Isn't eq. (3) essentially the same problem that you solve in QERA?
>
> Yes from a mathematical view, Caldera is a general case of QERA-exact. We did not realize this concurrent work throughout our research process. In the revised manuscript, we introduced Caldera in the related work section and used a whole section in the appendix to elaborate the connections between these two papers.
>
> However, they differ from a pratical use case. Caldera takes the form of $\mathbf{y}=\mathbf{x}(\widetilde{W}+A_{q,k} B_{q,k})$, where all weights and low-rank terms are quantized. QERA takes the form of $\mathbf{y}=\mathbf{x}(\widetilde{W}+A_{k} B_{k})$ where **only the weights are quantized**. For example, given an accuracy loss budget, we can expect QERA uses a smaller rank than Caldera as low-rank terms are in high-precision.
>
> > Additionally, the proof of QERA-Exact is not very different from Caldera. You are considering the symmetric square root of the auto-correlation matrix (or Hessian). If I understand correctly, this is essentially a simplification of the proof of Lemma 4.2 in Caldera under the assumption that the auto-correlation matrix is PSD. Please correct me if I'm mistaken?
>
> Sorry we cannot easily tell how [the proof of QERA-exact (Proof on page 5)](https://openreview.net/pdf?id=LB5cKhgOTu) is a simplification of [the proof of Lemma 4.2 in Caldera (Proof on page 16)](https://openreview.net/pdf?id=lkx3OpcqSZ) after comparing them. In the previous reply, we derived that the solution of QERA-exact is equivalent to the one in Caldera's Lemma 4.2, thus there must be some connection between the two proofs despite we cannot figure it out given the rebuttal time limitation. Also, one minor typo/mistake in your comments, autocorrelation matrix is always PSD, our Remark 1 assumes it is PD.

---

> ### Author Response · Authors · 2024-11-26
>
> # Answers to Follow-up Questions (Part 2)
>
> > Without the low-rank factors, isn't the optimization problem considered in your work the same as that in https://arxiv.org/abs/2004.10568? My point here pointing this out was merely the fact the minimizing the output error of a layer is not a very recent idea, and earlier references exist. See for example, the optimization problem of eq. (13) in https://arxiv.org/abs/2004.10568 exist. It is call it the task-based loss, but would you disagree that this is the same output error minimization formulation QERA considers? Please correct me again if I'm mistaken. I really appreciate the extensive experiments (including additional ones) that the authors did, but having a hard time distinguishing the novelty of the idea in the paper. Reiterating back, I would highly appreciate it if you could respond to them briefly without diving deep into details.
>
> We did not claim minimizing layer output error is new. We have restricted our claim in the problem of quantization error reconstruction, which takes the form of $\mathbf{y}=\mathbf{x}(\widetilde{W}+A_kB_k)$, where $\widetilde{W}$ is quantized, $A_k$ and $B_k$ are not quantized.
> - In this paper, we showed that for **the problem of quantization error reconstruction (QER)**, which takes the forms of $\mathbf{y}=\mathbf{x}(\widetilde{W}+A_kB_k)$, minimizing output error is more effective than minimizing weight approximation error, and we offer analytic solutions, QERA, to that problem.
> - The reason why we would like to address this is that [LoftQ](https://openreview.net/forum?id=LzPWWPAdY4), the oral paper in ICLR'24, focuses on minimizing weight approximation error. However we find LoftQ is not the effective way to solve this problem. Work in this domain is varied, with some research concentrating on weight error reconstruction (eg. [LRD (NeurIPS'21)](https://neurips2021-nlp.github.io/papers/27/CameraReady/Neurips_Workshop_camera_ready.pdf), [ZeroQuant-v2(NeurIPS'22)](https://arxiv.org/pdf/2303.08302v3), [LPAF (COLING'24)](https://aclanthology.org/2024.lrec-main.945/), [LoftQ (ICLR'24)](https://openreview.net/forum?id=LzPWWPAdY4)) and others on output error reconstruction. Currently, there is no consensus within the field of quantization regarding which type of error is the most advantageous to optimize. We show a head-to-head comparison between these via an empirical study to shed light on this issue. We use the model output error experiments in Fig.1 (line 324, page 7) and Fig.6 (line 918, page 18) to support our claim. We believe the observation that the decreased layer weight error in LoftQ leads to larger model output error is important but ignored by previous QER work.
>
> Again, we would like to iterate that **we do not want to state that we are the first and only to consider output error reconstruction, but would like to point out and ship the useful insights that performance wise this would work better than weight error reconstruction objectives for QER problem.**

---

> ### Author Response · Authors · 2024-11-26
>
> # Answers to Follow-up Questions (Part 3)
>
> > I really appreciate the extensive experiments (including additional ones) that the authors did, but having a hard time distinguishing the novelty of the idea in the paper.
>
> To help the reviwer have a better overview of our contributions, here we shortlist our contributions and **highlight** the advantageus ones even when compared to the concurrent work Caldera.
> - We formalize the quantization error reconstruction into two problems (Problem 1 and Problem 2 on page 4), and derives the solutions.
>   - The solution to Problem 1 is straightforward; we derives the exact solution to Problem 2, QERA-exact, though we just realized it agrees with the Lemma 4.2 in the concurrent work Caldera after the rebuttal started.
> - **We also offer an approximated solution, QERA-approx, which is much more computationally efficient than QERA-exact.**
> - We run extensive experiments to show the effecay of QERA-exact and QERA-approx; and have the following enlightening observations
>   - QERA-exact is suitable for post-training quantization, outperforming baselines by a clear margin. **It has more improvement over the baselines when the quantization is more aggressive.**
>   - **QERA-approx is suitable for fine-tuning (especially compared to QERA-exact/Caldera), outperforming baselines by a clear margin; besides, we also show that**
>     - **QERA-approx overcomes the pitfall in LoftQ: Reduced layer weight error $\ne$ reduced model output error (line 406, page 8). This pitfall is important but ignored by previous QER work.**
>     - **QERA-approx overcomes the pitfall in LQER: Model performance vs. calibration set size (line 469, page 9). This pitfall is also ignored by previous QER work.**
>     - **QERA-approx explains why the heuristic work in [LQER](https://openreview.net/forum?id=dh8k41g775) (line 317, page 6).**

---

> > ### Comment · Reviewer_8w8V · 2024-11-26
> > **Thank you and increasing my score**
> >
> > Thank you for your response.
> >
> > I appreciate the summary of contribution: QERA-approx, which proposes approximating the auto-correlation matrix as a diagonal matrix is computationally efficient. Although this is hypothesis is not very well-established in the paper, it shows reasonable performance -- not clear how this approximation would hold for other models.
> >
> > *Response to your answer (Part 1):*  *Sorry we cannot easily tell how the proof of QERA-exact (Proof on page 5) is a simplification of the proof of Lemma 4.2 in Caldera (Proof on page 16) after comparing them*
> >
> > See App. C.5 on Page 24 of the Caldera paper (https://arxiv.org/pdf/2405.18886). Except for reformulations, the proof is essentially equivalent for positive definite Hessians. I agree, it holds for PD, not PSD.
> >
> > *Response to your answer (Part 2):* I was saying minimizing output error (not weight error) has been used in prior works than those mentioned in the paper.
> >
> > Contingent on the fact that the authors consider the suggestions to place the work better in the context if existing literature, I believe the experimental contribution is appreciable, and I'll be increasing my score to 6.

---

### Meta-Review · Area_Chair_6pb3 · 2024-12-20

**Metareview:**

Dear Authors,

Thank you for your valuable contribution to the ICLR and the ML community. Your submitted paper has undergone a rigorous review process, and I have carefully read and considered the feedback provided by the reviewers.

This paper proposes a framework for analyzing the quantization error after it is compensated using low-rank, high-precision terms. The paper received overall positive scores (6,6,6,8,8). Given this positive assessment, I am willing to recommend the acceptance of your paper for publication. However, I urge you authors to carefully review Reviewer 8w8V's feedback again. On page 4 of the current paper, it is stated that "... to our best knowledge, there is no work providing an analytical solution to this problem.". However, the reference Saha et al. 2024 pointed out by the reviewer (which can be considered concurrent work) needs to be cited here since the theoretical result is near-identical (except a tiny difference in the objectives considered).

I would also like to remind you to carefully review all the other reviewer feedback and the resulting discussion. While most reviews were positive, the reviewers have offered valuable suggestions that can further strengthen the quality of the paper. Please take another careful look a the 'weaknesses' section of each reviewer comment. I encourage you to use this feedback to make any necessary improvements and refinements before submitting the final version of your paper.

Once again, thank you for submitting your work to ICLR.

Best,
Area Chair

**Additional Comments On Reviewer Discussion:**

The reviewer feedback helped clarify many issues raised by the reviewers. Reviewer 8w8V pointed out directly relevant papers. Some theoretical results partially overlap with Saha et al. 2024 which can be considered concurrent work and needs to be cited.

---

### Decision · Program_Chairs · 2025-01-22

Accept (Poster)